# Light-driven lattice soft microrobot with multimodal locomotion

Mingduo Zhang[1,5], Yuncheng Liu[1,5], Chunsan Deng [1], Xuhao Fan [1], Zexu Zhang[1], Shaoxi Shi[1], Fayu Chen[1], Huace Hu[1], Songyan Xue[1], Leimin Deng[1,2], Lige Liu[3], Tao Sun[3,4], Hui Gao[1,2] & Wei Xiong [1,2] ✉

Untethered microrobots hold significant promise in fields such as bionics, biomedicine, and micromechanics. However, replicating the diverse movements of natural microorganisms in artificial microrobots presents a considerable challenge. This paper introduces a laser-based approach that utilizes lattice metamaterials to enhance the deformability of hydrogel-based microrobots, resulting in untethered light-driven lattice soft microrobots (LSMR). Constructed from poly(N-isopropylacrylamide)-single-walled carbon nanotubes (PNIPAM-SWNT) hydrogels and a truncated octahedron lattice structure, the LSMR benefits from reduced relative density, which increases flexibility and accelerates light-driven deformation. By employing sequential laser scanning, the LSMR achieves various locomotion modes, including linear peristalsis, in situ rotation, and hopping, through adjustments in scanning frequency, trajectory, and laser power. The LSMR achieves a continuous in situ rotation speed of 29.38°/s, nearly 30 times faster than previous studies, and exhibits a peristaltic locomotion speed of 15.15 μm/s (0.14 body lengths per second). The LSMR can autonomously perform programmed motions under closed-loop feedback control and navigate through narrow openings as small as 75% of its resting width by actively deforming. Compared to a solid microrobot, the lattice microrobot requires only one-sixth of the laser energy to achieve three times the motion speed, under otherwise identical conditions. These advancements mark a significant leap forward in the design and functionality of light-driven soft microrobots, offering promising avenues for future research in biomedicine, bionics, and micromechanical engineering.

Microscale robotics has gained significant attention due to its potential applications in biomedicine, environmental monitoring, and precision micromechanics[1–3]. Among them, submillimeter soft microrobots are particularly promising due to their small size, adaptability, and diverse actuation possibilities[4–8]. Biological submillimeter microorganisms have evolved highly efficient and adaptable movement strategies to navigate complex environments[9,10]. Inspired by these mechanisms, bioinspired motion patterns such as peristalsis[11,12], inchworm-like crawling[13,14], walking[15–17], and swimming[18,19] have been applied to the design of soft microrobots, expanding the range of achievable

[1]Wuhan National Laboratory for Optoelectronics, School of Optical and Electronic Information, Huazhong University of Science and Technology, Wuhan 430074, China. [2]Optics Valley Laboratory, Wuhan 430074, China. [3]State Key Laboratory of High End Heavy Load Robots, Foshan 528300, China. [4]Artificial Intelligence Research Center, Midea Group, Shanghai 201702, China. [5]These authors contributed equally: Mingduo Zhang, Yuncheng Liu. ✉e-mail: weixiong@hust.edu.cn

locomotion strategies[20–22]. However, achieving controlled, efficient and adaptive motion in such small systems remains a considerable challenge. The constraints in actuation efficiency and limited degrees of freedom continue to hinder the realization of highly versatile and multimodal locomotion. The integration of advanced actuation materials, precise actuation strategies and bioinspired structural designs offers a potential solution to improve the locomotion efficiency and adaptability.

One of the main constraints on microrobot performance is the choice of actuation materials and structural design. Hydrogels are commonly employed due to their biocompatibility, tissue-like mechanical properties, and various actuation modalities, including magnetic fields[23–25], pH[26,27], humidity[28–30], temperature[31] and light[11,32–34]. However, the slow fluid movement within hydrogels reduces their deformation speed, limiting actuation efficiency. Typically, optimizing material composition is a common approach to enhancing hydrogel performance. On the other hand, designing artificial microstructures, such as lattices, has also been found to improve the response speed and deformation range of hydrogels[33,35].

Meanwhile, structural design plays a crucial role in shaping the motion strategies of soft microrobots. Many existing approaches rely on predefined periodic deformations to achieve movement, demonstrating relatively rapid locomotion in controlled environments[13–15,36]. However, these methods often encounter challenges in complex surroundings, where the ability to reconfigure and adapt to external stimuli is essential for effective navigation. One approach to address this challenge is to enable localized deformations throughout a fully soft microrobot, allowing different regions of the body to undergo shape changes in a programmable manner without being constrained to a single predefined deformation mode. Leveraging the high spatial resolution of light stimulation, precise control over localized shape transitions can be achieved, supporting multimodal motion. By integrating a fully soft body structure with precision light actuation, this design improves adaptability in complex terrains and enhances control over deformation and movement, enabling microrobots to operate more effectively in diverse environments.

To address the limitations above, we present a light-driven lattice soft microrobot (LSMR) constructed from poly(N-isopropylacrylamide)-single-walled carbon nanotubes (PNIPAM-SWNT) hydrogels with a truncated octahedron lattice structure to enhance deformability and actuation efficiency. The lattice design reduces relative density, allowing for greater flexibility and faster deformation under light stimulation compared to solid microrobots made from the same hydrogel precursor material. The LSMR generates localized body deformations under high-precision laser stimulation, enabling fine control over shape transformation. By adjusting scanning frequency, trajectory and power, these deformations generate different locomotion modes, including linear peristalsis, in situ rotation and hopping. The LSMR exhibits a peristaltic locomotion speed of 15.15 μm/s (0.14 body lengths per second) and an in situ rotation speed of 29.38°/s. Compared to solid microrobot, the lattice microrobot requires only one-sixth of the laser energy to achieve three times the motion speed, under otherwise identical conditions. The LSMR can squeeze through constrictions as narrow as 75% of its resting width under laser actuation and autonomously follows programmed paths with closed-loop feedback. These advancements demonstrate that the integration of lattice-based structural design and precise light-driven actuation contributes to improved energy conversion efficiency, facilitates multimodal locomotion, and enhances adaptability in constrained environments, which are critical challenges in the development of soft microrobots. The integration strategy may support future applications in areas such as high-precision multi-agent control, high drug-loading capacity for controlled release.

## Results

### Design and manufacturing of lattice structure

The lattice structure was fabricated using laser direct writing (LDW), a precise 3D micro-additive manufacturing technology commonly used in microrobotics[37,38]. The device and method used for fabricating the lattice structure are depicted in Fig. 1a. Building on our previous research[33], we selected N-isopropylacrylamide (NIPAM) as the monomer, a temperature-responsive material. Single-walled carbon nanotubes (SWNTs) were incorporated to enhance the photothermal conversion effect. To ensure easy detachment and transfer of the LSMR for application, a dextran thin-film sacrificial layer was spin-coated onto the $SiO_2$ substrate[39], as depicted in Fig. 1b. This dextran layer remains intact during development but dissolves in water, allowing for smooth release. A 0.5% (v/v) Tween-20 aqueous solution served as the medium for releasing and operating the LSMR. As a surfactant, Tween-20 reduces adhesion between the LSMR and the substrate, as well as between individual microrobots[11]. Once released, the LSMR was transferred to the application environment using a pipette and activated by laser, as illustrated in Fig. 1c. A complete overview of the process is provided in Supplementary Fig. S1.

The LSMR is composed of truncated octahedron microstructures, a common lightweight and high-strength lattice metamaterial[40]. The model and scanning electron microscope (SEM) images are presented in Fig. 1d, e. The truncated octahedron design combines hexagonal and tetragonal cells, constructed using rods of uniform length (L) and diameter (D). Cells with specific parameters (rod length L of x μm, rod diameter D of y μm, LxDy) are joined by quadrilateral faces to form a block, as shown in Fig. 2a. By varying the rod dimensions, we produced lattice structures with different relative densities and porosities. Additional metamaterial microstructures were also created and are displayed in Supplementary Fig. S2.

### Light-driven deformation of lattice structure

To investigate the impact of structural parameters on deformation, we measured the edge lengths of various lattice structures in both swollen and shrunken states (Supplementary Fig. S3; design parameters in Supplementary Table 1). The base of each block adhered to the substrate, limiting its shrinkage, while the apex contracted with minimal constraints. This caused the rectangular blocks to form a truncated square pyramid upon shrinking. The simulation results are consistent with the experimental results, proving that our material model can well fit the deformation properties of lattice-structured hydrogel materials (Fig. 2b, c). We measured the edge lengths in both the swollen ($s_d$) and shrunken ($s_s$) states at the apex of each block and defined the shrinkage ratio ($\varepsilon$) to quantify the deformation of the lattice structure.

$$\varepsilon = \frac{s_d - s_s}{s_d} \tag{1}$$

As the relative density of the lattice structure decreases, the shrinkage ratio correspondingly increases (Fig. 2d). In our experiments, the average linear shrinkage ratio of the solid structure is 15.08%, while the lattice structure achieves an average maximum shrinkage ratio of approximately 39.90% (L2D2.5, L4D4, L6D5). This reflects an enhancement of approximately 25% in the deformation capacity of the lattice structure under identical process parameters. It is also noted that this enhancement has a limit. The relative density of the lattice structure decreases as the rod diameter of the truncated octahedron is reduced. When the diameter is smaller than the rod length, the rods are not strong enough due to the current process parameters, which ultimately leads to lattice collapse and prevents the formation of a complete three-dimensional structure. For this reason, the L2 series in the figure contains only five data points, in contrast to the other series.

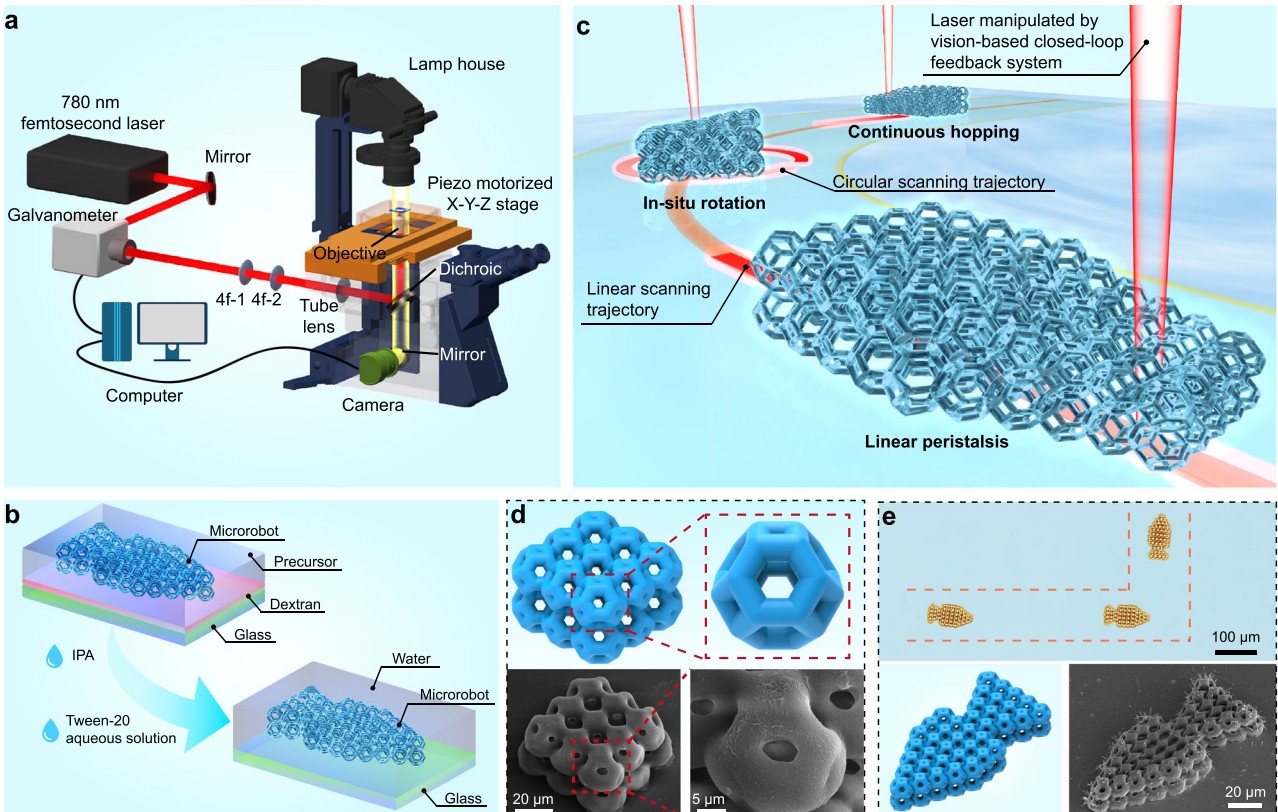

**Fig. 1 | Design and manufacturing of LSMR. a** Schematic illustrating the system of the lattice microstructure LDW fabrication and vision-based closed-loop feedback system. **b** Schematic diagram of the process of releasing the microrobot after LDW fabrication. **c** The schematic diagram illustrates the LSMR in the application scenario of multimodal locomotion. **d** 3D model and SEM image of truncated octahedral lattice structures (rod length L of 8 μm, rod diameter D of 8 μm) arrays and image of individual microstructures. **e** Optical micrograph, 3D model, and SEM image of truncated octahedrons (rod length L of 4 μm, rod diameter D of 4 μm) composed of bionic fish shapes in LSMR.

Additionally, we observed that the shrinkage ratio decreased with increasing rod diameter at comparable relative densities. This difference is attributed to the varying cross-link density within the rods[33]. During LDW, the polymerized region expands with radical diffusion, leading to mild polymerization at the center when the laser spot is focused on the rod's edge. As the laser spot moves toward the core of the rod, the polymerization intensifies, creating a higher degree of cross-linking in the rod's core. This central region experiences less deformation, while the less cross-linked outer region shows greater deformation.

We further examined the photoresponsive properties of the lattice structures by measuring deformation magnitude and response time under the same light field irradiation (Supplementary Movie 1). The lattice structure achieved 30.58% (95,176 μm³) volume shrinkage in just 1.46 s, whereas the solid structure took 2.50 s to shrink by only 10.55% (22,788 μm³) (Fig. 2e). The shrinkage rate of the lattice structure was 7.15 times faster than the solid structure.

The faster response of the lattice structure is due to its larger surface area and hollow architecture, which reduces the distance solvent molecules must travel to exit the hydrogel. As the temperature reaches the phase transition of PNIPAM, water molecules flow from the hydrogel interior to the surrounding environment under pressure and chemical potential gradients[41,42]. The shorter diffusion distance in the lattice structure allows for faster water expulsion compared to the solid structure.

We measured the stiffness of individual blocks using a micro mechanical testing system. The stiffness of blocks with rod lengths of 6 μm decreased as relative density decreased (Fig. 2f). The experimental setup and measurement process are detailed in Supplementary Fig. S4,

while additional stiffness data for individual blocks are provided in Supplementary Fig. S5.

## Linear peristalsis by scanning frequency modulation

Inspired by the sequential contraction and deformation observed in the unicellular microalga *Euglena gracilis*[43,44], we developed a peristaltic propulsion mode for the LSMR. Based on the evaluation of structural stability, deformation performance, and scalability, we selected L4D4 as the unit structure for the LSMR. Among the candidate structures (L2D2.5, L4D4, and L6D5), all demonstrated significant shrinkage ratios under external stimuli; however, L2D2.5 exhibited limitations in scalability due to its near-closed cavities and reduced porosity, which hindered solvent exchange and photothermal responsiveness at larger scales. While both L4D4 and L6D5 provided sufficient porosity for efficient deformation, L4D4 showed significantly higher stiffness (~250 μN/μm compared to ~75 μN/μm for L6D5), ensuring better structural integrity during dynamic operations. This balanced design of L4D4 achieves optimal performance in terms of deformation capability and mechanical robustness, making it the most suitable choice for LSMR applications.

Mimicking *Euglena* locomotion, the LSMR undergoes a six-phase movement cycle, as shown in Fig. 3a (Supplementary Movie 2). The laser sequentially induces localized shrinkage and swelling in the microrobot, creating ordered deformations that redistribute frictional forces between the microrobot and its substrate. Areas undergoing swelling generate higher friction than those shrinking, as the lattice structure contracts along the Z-axis, adjusting the microrobot's contact points with the environment. Due to the sequential scanning of the laser, there is a difference in the maximum shrinkage time between the

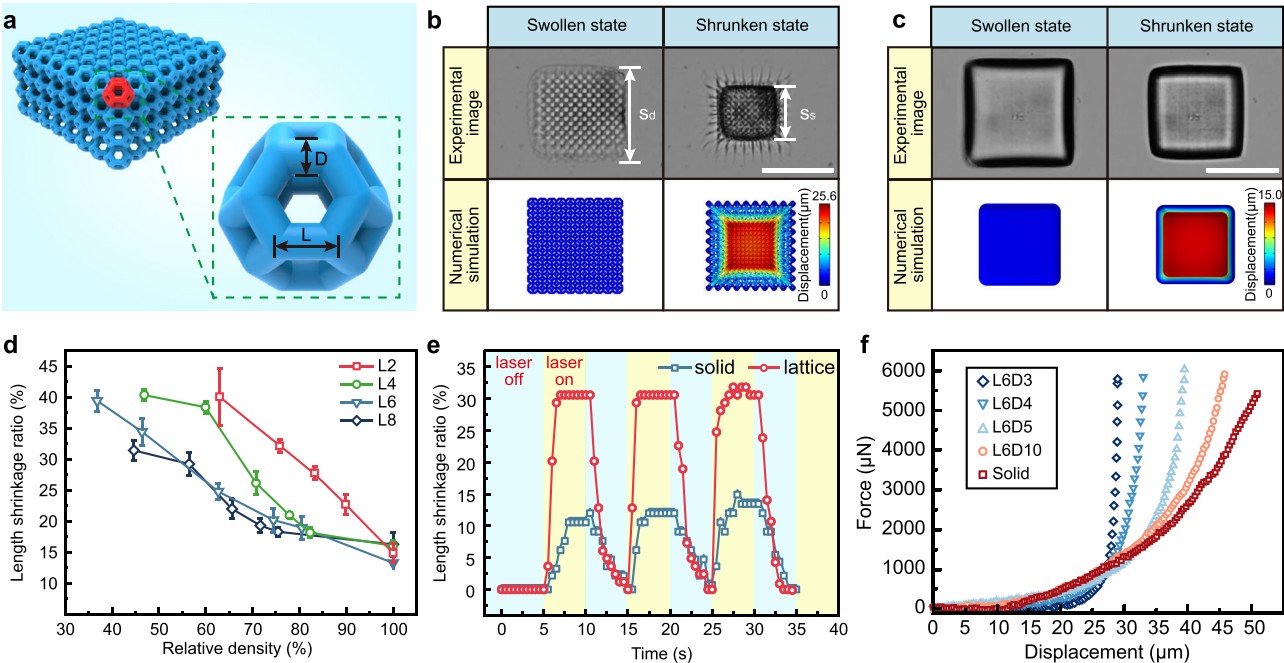

**Fig. 2 | Design and light response of LSMR. a** Schematic illustrating the truncated octahedral lattice metamaterial microstructure array model and the monolithic model. **b** Optical microscopy images and finite element simulations of the lattice structure (L2D2) in the swollen and light-driven shrinking states. Scale bar: 50 μm. **c** Optical microscopy images and finite element simulations of the solid structure in the swollen and light-driven shrinking states. Scale bar: 50 μm. **d** Shrinkage ratios for lattice structures with different parameters. Data points with the same color and symbol represent unit microstructures with the same rod length L. Specifically, red squares, green circles, cyan downward triangles, and dark blue diamonds correspond to rod lengths of 2 μm, 4 μm, 6 μm, and 8 μm, respectively. Error bars represent standard deviation. **e** Comparison of light-driven deformation speed between lattice and solid structures. The blue background means the laser is off, and the yellow background means the laser is on. **f** Force-displacement curves for lattice and solid structures with the rod length of 6 μm.

front and end of the LSMR, with the end shrinking earlier than the front in each scan cycle (Fig. 3b).

In a single cycle, the microrobot's step displacement is determined by the difference between its forward displacement (from state i to state iii) and its backward displacement (from state iii to state iv). Figure 3c details the displacement within a single scan cycle. The microrobot's step displacement is determined by the difference between its forward displacement (from state i to state iii) and backward displacement (from state iii to state iv). The LSMR's speed is derived from the step displacement and scanning frequency, expressed in Eq. (2) (complete derivation in Supplementary Note S1).

$$v_l = L_b\left[\left(A_2 + \frac{A_1 - A_2}{1 + e^{d/v_s - x_0/dx}}\right) - \left(y_0 + A\left(\frac{frac}{1 + e^{d/v_s - x_{01}/k_1}} + \frac{1 - frac}{1 + e^{d/v_s - x_{02}/k_2}}\right)\right)\right]\frac{v_s}{L_s}$$
(2)

In this equation, $L_b$ represents the diameter of the heat affected region in the body length, $d$ is the distance affected by laser heating at the robot's end, $L_s$ is the length of a single laser scanning path and the coefficients in Eq. 2 were determined by fitting the photothermal shrinkage and swelling process of the LSMR observed in Fig. 2e. A detailed discussion of the parameter sources and their physical implications is provided in Supplementary Note S1. We found that the velocity ($v_l$) of the LSMR primarily depends on the laser scanning frequency($v_s/L_s$), which initially increases with $v_s$, then decreases, consistent with related studies[14,45]. The scan length ($L_s$) is typically chosen based on the intended movement distance. For longer peristalsis motions, we control the laser to perform repeated unidirectional scans between designated start and end points. Experimental results showed that the LSMR achieved a maximum speed of 15.15 μm/s (0.14 body length/s) at scanning speeds between 1600–2200 mm/s and the scanning length is 160 μm (Supplementary Fig. S6). The

optimal laser scanning frequency is in the range of 10–14 Hz (60 mW). This variation in speed is primarily due to the inverse relationship between step distance and the number of scans (Supplementary Fig. S7). As scanning speed increases, the peristaltic motion accelerates until reaching a peak, after which speed declines due to reduced deformation until no movement occurs.

Laser power directly affects the selection of scanning frequency. At lower laser powers, the LSMR does not deform sufficiently, so scanning speed must be reduced to allow for heat accumulation. For example, at 30 mW, the maximum speed achieved is 2.22 μm/s at a scanning frequency of 4.65 Hz, with nearly no movement at 10 Hz. Higher laser powers demand faster scanning speeds to prevent the LSMR from floating due to strong thermal effects. However, even when scanning speed is increased, excessive heating results in overall contraction and dissolution of the LSMR rather than precise ordered deformation. For instance, at 80 mW, a frequency below 3 Hz causes floating, while speeds of 10–14 Hz yield only 9 μm/s. Based on these observations, we determined that 50–60 mW provides the optimal laser power for effective motion. In our model, body length ($L_b$) does not significantly impact movement speed. For effective motion, it is sufficient that the body length exceeds the diameter of the laser's heat-affected zone on the LSMR, allowing for controlled, localized deformation. Ideally, only the heat-affected zone deforms, shifting along the LSMR as the laser scans. Extending body length without adjusting the heat-affected zone size does not increase the speed. However, if the body is too short, the entire LSMR deforms rather than producing targeted, orderly deformation. Based on this, we selected a body size of approximately 100 × 50 × 20 μm for our experiments.

To compare the movement speeds of solid and lattice structures under the same scanning speed (120 μm/s), both structures were designed with identical three-dimensional dimensions to ensure a fair comparison. Since they were fabricated under the same femtosecond

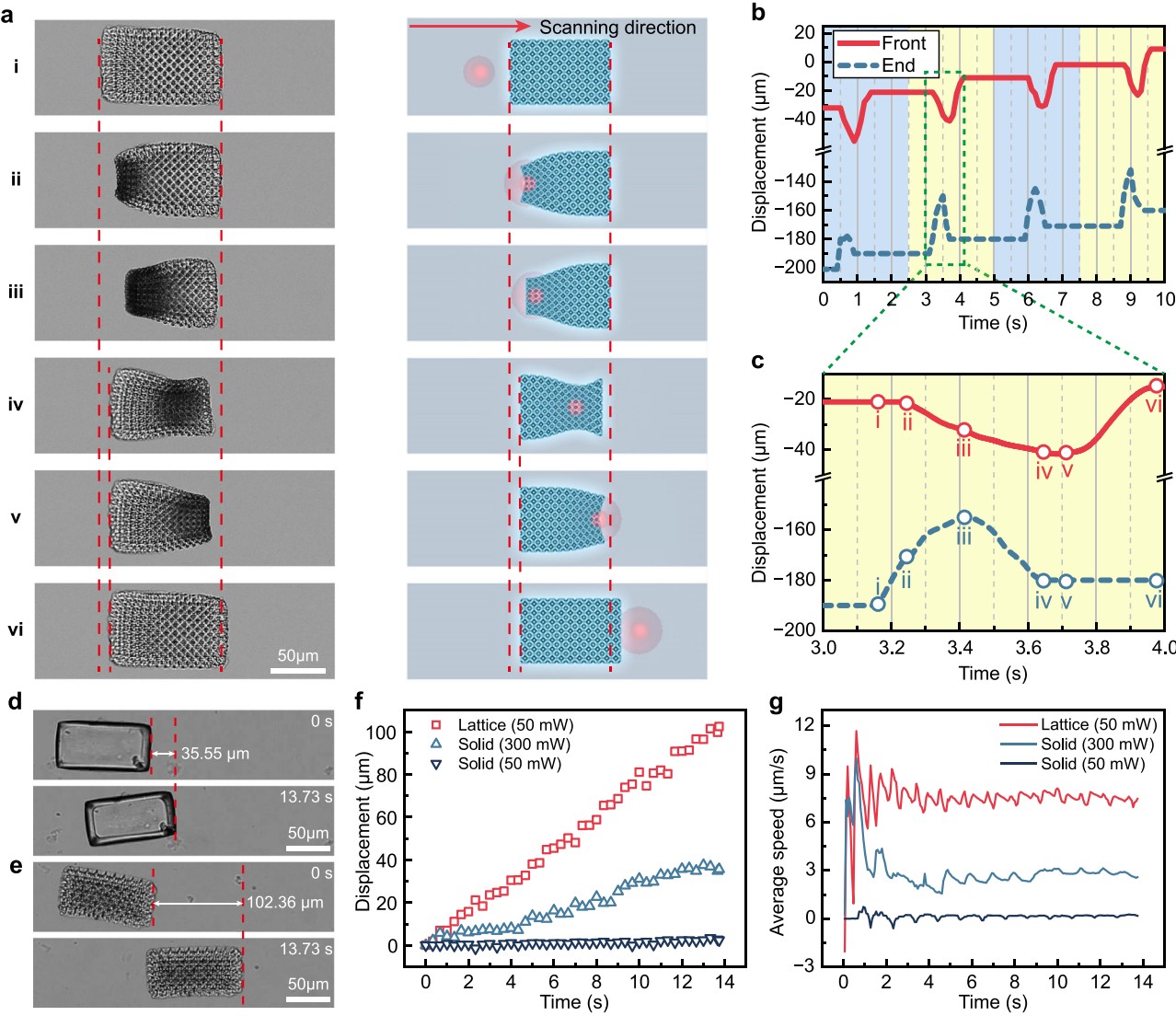

**Fig. 3 | Linear peristalsis of LSMR under laser scanning frequency modulation. a** Optical microscopy images of scanning light-driven LSMR linear peristalsis (top view) and schematic illustrating the relative position of the laser (side view). **b** Front and end displacements in four laser scanning cycles. The blue background and yellow background in the figure represent one laser scanning cycle each. **c** Displacement of the front and end of the LSMR within a single laser scanning cycle. **d**, **e** Experimental images of light-driven linear peristalsis of the solid robot (300 mW) and lattice robot (50 mW). **f** Displacement of solid microrobot and lattice microrobot. **g** The average speed of solid microrobot and lattice microrobot. The initial 2 s is excluded from the analysis due to the jitter.

laser direct writing parameters, their movement behaviors could be directly evaluated; the lattice design inherently reduces weight, which is one factor in the improved agility and efficiency observed. The solid structure (laser power 300 mW) and the lattice structure (laser power 50 mW) traveled different distances over 14 s (Fig. 3d, e). (Supplementary Movie 3). The solid structure moved at 2.59 μm/s, while the lattice structure moved at 7.45 μm/s (Fig. 3f, g). When driven at 50 mW (15.82 mJ/cm²), the solid structure moved at just 0.18 μm/s, remaining nearly stationary (Fig. 3g). Conversely, at 300 mW (94.91 mJ/cm²), the lattice structure floated upwards rather than performing peristalsis, due to the Marangoni effect and photothermal buoyancy from the excessive heat[46,47].

## Motion manipulation by laser scanning trajectory

To investigate the impact of laser scanning trajectories on the locomotion of the laser-steered microrobot (LSMR), we conducted experiments using both linear and circular scanning methods. A linear laser scanning path aligned with the LSMR's body orientation (power: 60 mW, scanning speed: 120 μm/s) was applied, enabling controlled

movement (Fig. 4a) (Supplementary Movie 4). The superimposed image of the LSMR's motion highlights its trajectory (Fig. 4b), while position and timing data in the X and Y directions reveal an average speed of 5.25 μm/s (Fig. 4c).

By adjusting the laser scanning trajectory offset relative to the microrobot's geometric center, we could finely tune the direction of the LSMR's locomotion. The focused laser allowed for localized deformations rather than uniform changes across the body, creating asymmetrical deformation that facilitated controlled steering. Experimental results demonstrated a maximum angular displacement of 16°, illustrating the effectiveness of laser-guided actuation for precise directional control.

For rotational control in the X-Y plane, we employed a circular scanning trajectory tailored to the microrobot's dimensions (Supplementary Movie 4). The trajectory induced controlled rotational motion through localized deformations in a circular pattern (Fig. 4d). The laser rotates clockwise around the LSMR's center of mass, generating a rightward peristalsis at the front and a leftward peristalsis effect at the end of the LSMR (Fig. 4e). This rotation creates a rightward peristalsis

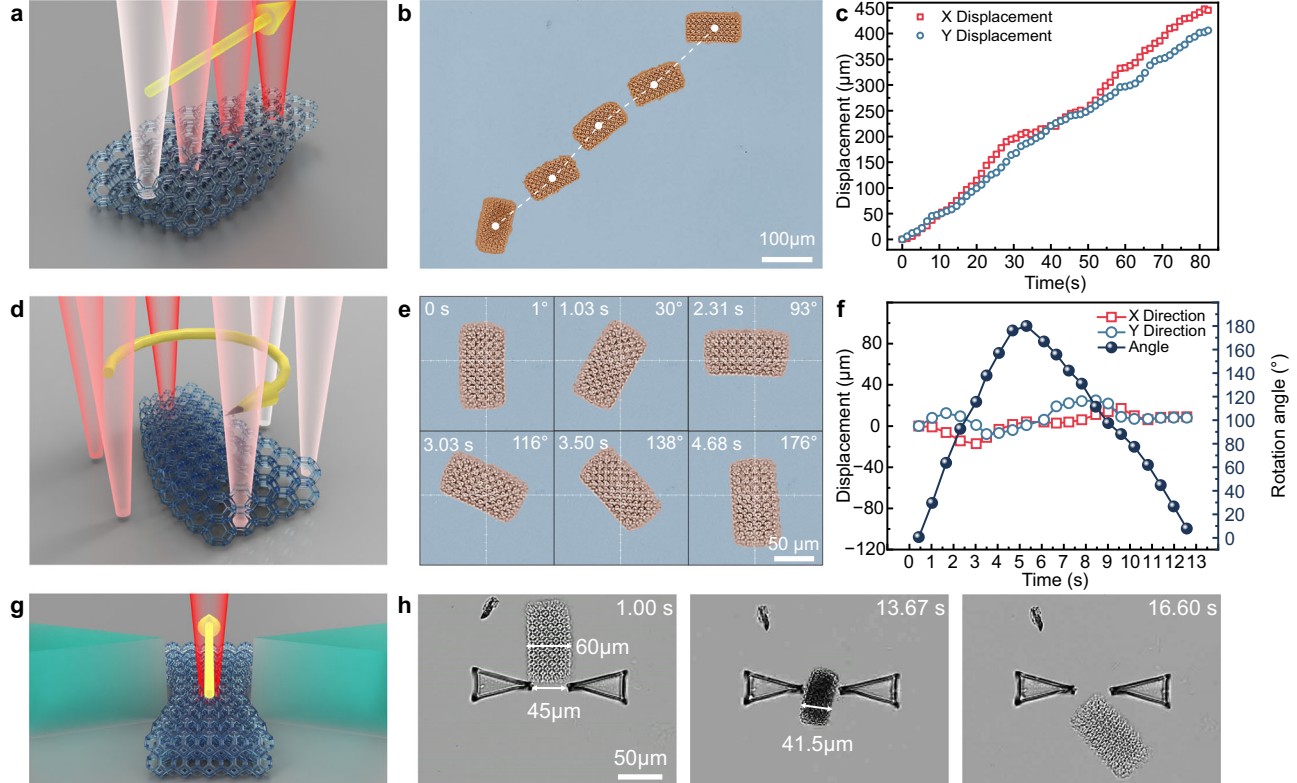

**Fig. 4 | Planar composite movement of LSMR modulated by laser trajectory.**
**a** Schematic of the linear peristalsis. Gradual color change of the light path (from white to red) represents the laser scanning trajectory. The yellow arrow represents the direction of laser scanning. **b** Superimposed images of continuous linear peristalsis. **c** Displacement of continuous linear peristalsis at different time points. **d** Schematic of laser-actuated region for in situ rotation. **e** Sequential optical images of clockwise rotation processes. **f** Rotation angle over time and center-point offset in X and Y directions during rotation. **g** Schematic of squeezing through the slit. The red fluorescent line represents the scanning path of the laser. **h** Sequential optical images of the LSMR squeezing through the slit by active body deformation.

at the front of the LSMR and a leftward peristalsis effect at the end of the LSMR. The combined effect generates a clockwise torque across the entire LSMR. Conversely, counterclockwise scanning of the laser produces a corresponding counterclockwise torque. By adjusting the direction of peristalsis at specific points along the LSMR's body, the overall direction of the robot's movement can be controlled.

Angular variations and center position offsets throughout the ±180° scanning process were recorded, with an average rotational speed of 29.38°/s over 360° and a maximum speed of 51.02°/s (Fig. 4f). The speed exceeds previously reported rotation speeds of 1°/s for planar peristalsis[12,48,49], showcasing the efficiency of our laser-induced mechanism for rapid, controllable motion. The difference in rotational speeds during clockwise and counterclockwise rotations can be attributed to the eccentricity of the scanning path relative to the LSMR's center. When the circular trajectory deviates from the microrobot's structural center, the misalignment reduces the efficiency of induced deformations, diminishing the rotational speed. Additionally, this deviation introduces translational motion in the X and Y axes alongside intended rotation, as depicted in Fig. 4f, where the maximum observed offset was 17 μm, about 15% of the LSMR's total body length. This underscores the need for precise alignment to optimize rotational performance and minimize unintended lateral movement.

To quantify the angular precision of the LSMR, we conducted experiments using laser powers of 20 mW, 30 mW, 40 mW, and 50 mW at a scanning speed of 120 μm/s (Supplementary Fig. S8). The angular displacement per complete circular laser scan increased with laser power, averaging 3.87° ± 1.49°, 6.21° ± 2.80°, and 6.59° ± 3.62°, respectively. No rotation was observed at 20 mW, as the thermal input was insufficient to induce effective deformation of the hydrogel structure, resulting in a lack of actuation force and thus no measurable

rotation angular. At 30 mW, the LSMR began to rotate, though the motion remained unstable and was easily impeded by surface irregularities due to the limited deformation and actuation force. At 40 mW, a more robust and consistent rotation was achieved, suggesting that the deformation had reached an effective threshold. Further increasing the power to 50 mW resulted in only a slight gain in rotation angle but introduced larger variability, likely due to the expanded thermal influence zone and less localized actuation. These results demonstrate that rotational behavior is tunable by laser power, with a trade-off between actuation intensity and motion stability that must be optimized for precise path-following applications.

To demonstrate the LSMR's soft and adaptable capabilities, we designed it to squeeze through a narrow slit of 45 μm, smaller than its 60 μm body width (Fig. 4g) (Supplementary Movie 5). The LSMR's volume contraction upon laser irradiation allowed the LSMR to navigate through the narrower gap (Fig. 4h). Theoretically, the smallest gap an LSMR can navigate is equivalent to its contracted body width, assuming it relies solely on frictional interactions with its environment. In practice, however, achieving this limit is challenging, as the microrobot undergoes repeated contraction-dissolution peristalsis while passing through a slit. During dissolution, lateral forces from the slit act on the microrobot's body, which can destabilize and deform it laterally. This destabilization depends on the microrobot's height-width ratio and the Young's modulus of its material. If this deformation continued, the microrobot may eventually detach from the substrate or even topple, as shown in Supplementary Fig. S9. This is, to our knowledge, the first instance of a light-driven micrometer-scale soft-bodied robot squeezing through a slit narrower than its body width, relying solely on body-environment interactions rather than external assistance[50] or passive deformation[51,52].

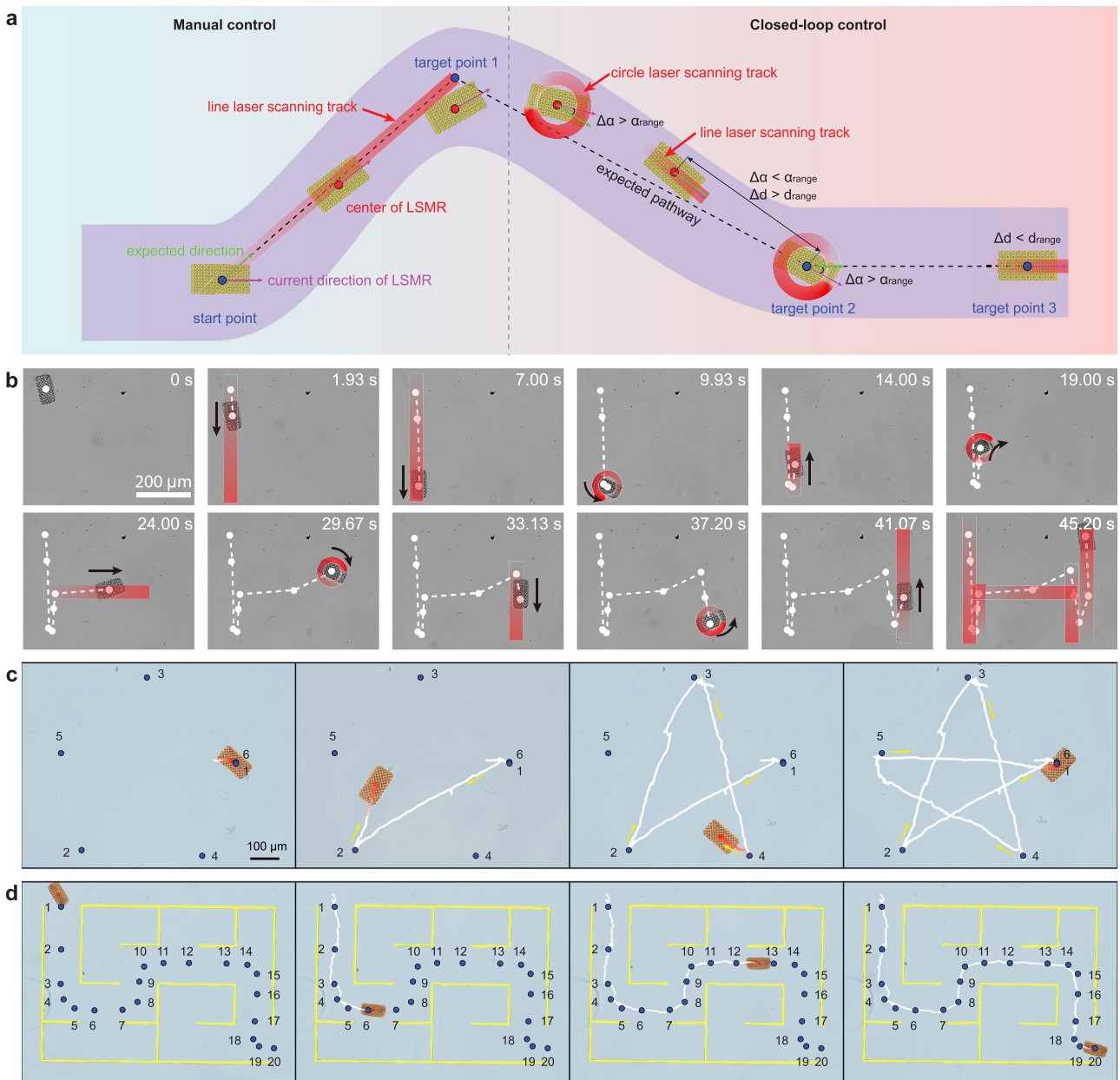

**Fig. 5 | Closed-loop control of the peristalsis. a** Schematic of manual and closed-loop control. **b** Microscopy image sequences of H-path motion under manual control, with red trace indicating laser scanning path. **c** Microscopy image sequences of pentagram path motion under closed-loop control. The white line represents the LSMR trajectory. Blue dots represent target points, and the numbers represent target point sequences. **d** Microscopy image sequences of maze path motion under closed-loop control. The area of the maze is marked in yellow. Scale bar: 200 μm.

## Closed-loop control of the peristalsis direction and destination

The LSMR leverages a synergistic combination of linear peristalsis and rotational motion to perform complex programmed movements on a substrate. Precise trajectory control is enhanced through closed-loop feedback mechanisms and computer vision recognition systems. The difference between open-loop and closed-loop control for peristalsis shown in Fig. 5a. In the manual control scenario (left), the laser follows a predetermined path, while the closed-loop scheme (right) employs real-time feedback to assess the LSMR's angular orientation and distance to the target after each drive cycle. The program flowchart for the closed-loop control system is provided in Supplementary Fig. S10.

To demonstrate manual control, we designed an "H" shaped path (Supplementary Movie 6). Initially, the laser scanning trajectory was set, and the piezoelectric stage was adjusted to position the LSMR at the start of the scanning path. This process was repeated at each inflection point of the path. Despite these efforts, the LSMR exhibited directional deviations (Fig. 5b). Occasional manual adjustments of the piezoelectric stage were necessary to keep the LSMR within the laser scanning trajectory during linear peristalsis.

We further validated the effectiveness of the closed-loop drive mechanism by designing two trajectory patterns: the "Pentagram" and the "Maze" (Fig. 5c, d) (Supplementary Movie 7). Unlike manual control, this system does not require precise initial alignment with the laser scanning path. After setting the target points, the system autonomously guides the LSMR to each point using a negative feedback loop. This loop focuses on the angular deviation between the LSMR's current orientation and the target direction. If the deviation exceeds a set threshold, the system

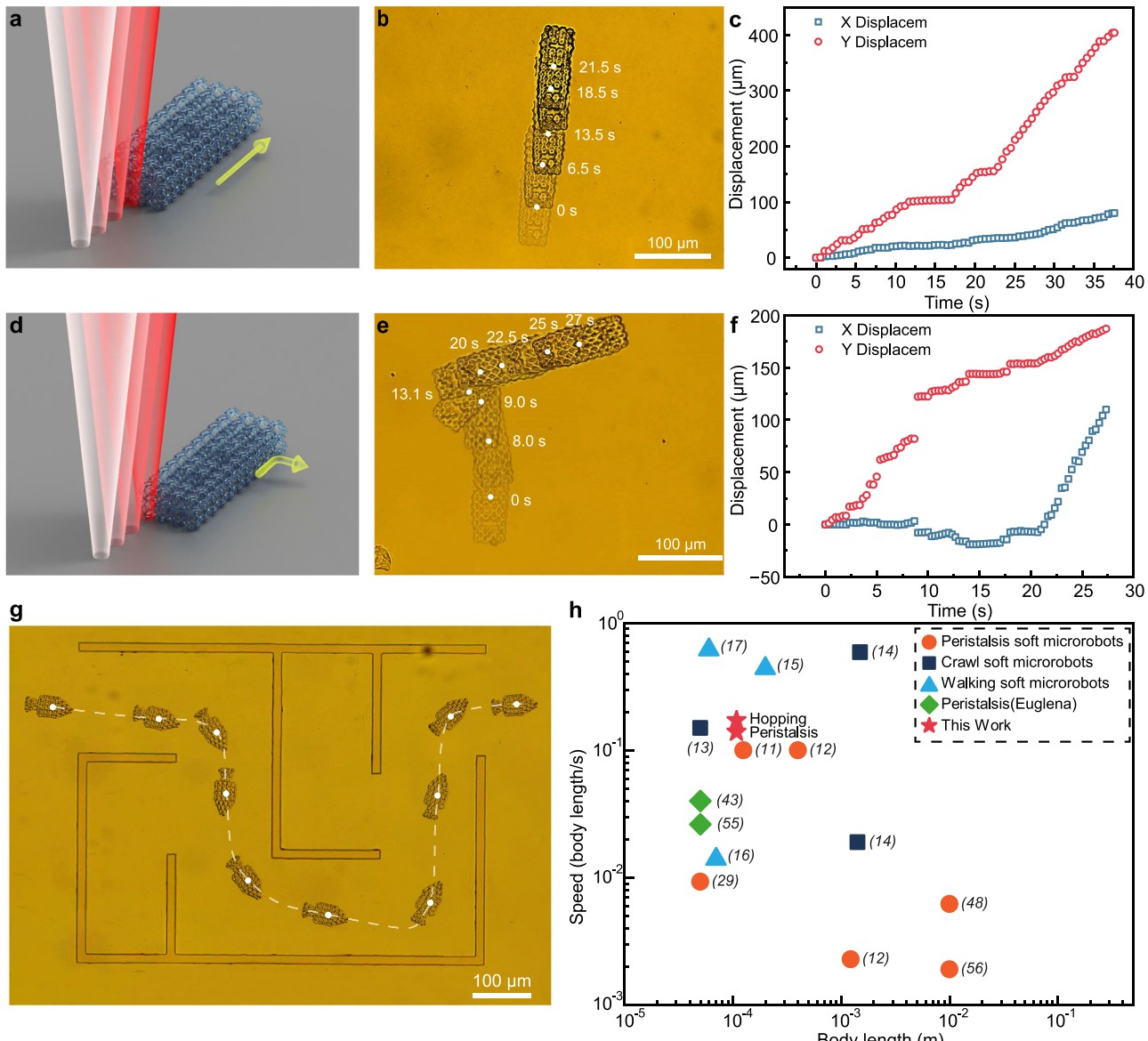

**Fig. 6 | High-power laser-driven LSMR continuous hopping. a** Schematic illustration of the laser-driven LSMR performing linear continuous hopping. **b** Superimposed images of continuous linear hopping and (**c**) displacement with respect to time. **d** Schematic of laser action region for right-turn continuous hopping. **e** Superimposed images of right-turn hopping and (**f**) displacement with respect to time. **g** The stitched movement path of the fish-shaped LSMR maneuvering through the maze driven by continuous hopping. **h** Summary of reported robots plotted as the ratio of locomotion speed to body length as a function of length.

employs circular scanning to adjust the angle before switching to linear scanning to reach the target.

Continuous feedback from real-time images captured by a microscope camera updates the LSMR's position and orientation after each scan. Detailed angle determination methods are provided in Supplementary Fig. S11. The use of redundant waypoint annotations allows for more precise control of the LSMR's trajectory, reducing deviations caused by environmental and system precision factors, as demonstrated in Fig. 5d. For stable movement, the laser power is set to 25 mW, with scanning speeds of 800 µm/s for linear paths and 2000 µm/s for circular paths. This approach allows the system to autonomously adjust its trajectory in real-time, facilitating precise control in complex environments.

**Continuous hopping under high laser powers**

The multimodal locomotion of the LSMR was further investigated under high laser power conditions[53,54]. Thermophoresis, a

phenomenon where particles in a solution move along a temperature gradient, was utilized to achieve stable forward hopping by increasing the laser power to 200 mW. Continuous hopping was facilitated by adjusting the piezo stage to apply the laser scan to the LSMR.

The high-power laser, directed at the tail end of the LSMR's body, creates a temperature gradient in the solution that drives forward movement during linear hopping (Fig. 6a). A sequence of images from the linear hopping experiments demonstrates this behavior (Fig. 6b, Supplementary Fig. S12, Supplementary Movie 8). Variations in the laser's action area led to deviations from the intended movement path, with displacement observed in both the X and Y directions. The average movement speed of the LSMR was 9.36 µm/s, with a maximum speed of 20.63 µm/s (Fig. 6c). While increasing the movement speed of the piezo stage could enhance the LSMR's speed, it also led to greater path deviations.

Compared to the peristaltic motion as shown in Fig. 3e, the hopping LSMR demonstrates only 39.46% of the energy conversion

efficiency of the peristaltic LSMR. Although the hopping LSMR achieves a speed 1.26 times that of the peristaltic LSMR, it requires four times the laser energy. This efficiency gap arises because, during hopping, the laser primarily acts on the microrobot's tail and only intermittently affects the main body. Despite continuous laser scanning, a significant portion of the laser's active time does not contribute directly to driving the hopping motion.

Adjusting the laser's action area can alter the forward direction of the LSMR. When the laser is positioned to the left of the body axis, it generates a clockwise torque alongside the forward thrust, causing the LSMR to curve forward toward the right front (Fig. 6d). A sequence of images captures the right-turn hopping process (Fig. 6e, Supplementary Fig. S12, Supplementary Movie 8). During this maneuver, the LSMR achieved a rotation angle of 73.89°, with an average rotation speed of 14.76°/s and a maximum speed of 39.67°/s, as shown by the displacement in the X and Y directions (Fig. 6f).

To demonstrate continuous hopping, we navigated the LSMR through a maze photolithographically patterned onto the substrate using UV-cured AZ-5214 photoresist, featuring four designated turn areas along the navigation path (Fig. 6g). We guided the bionic fish-shaped LSMR to continuously hop through the maze by controlling the laser field and scanning direction (Supplementary Movie 9), showcasing the microrobots' potential for applications in complex environments.

Finally, the body length and movement speed of various soft-bodied robots with peristalsis locomotion are compared (Fig. 6h and Supplementary Fig. S13). The LSMR achieves a peristalsis speed comparable to that of natural Euglenoids[55]. While further increases in LSMR movement speed are possible through higher laser power and scanning speeds, these adjustments often lead to uncontrollable movements in experimental settings. Alternatively, the use of spatially anisotropic hydrogel architectures in combination with photothermal actuation has been reported to offer a potential means of improving the controllability of peristaltic motion[56].

## Discussion

In this work, we developed a light-driven lattice soft microrobot (LSMR) based on truncated octahedral mechanical metamaterials to achieve multimodal bionic locomotion. The lattice architecture reduces the hydrogel's relative density, enhancing the magnitude and speed of light-driven deformation. Through sequential laser heating, the LSMR realizes peristalsis, in situ rotation, and hopping motions, while a closed-loop optical feedback system automates trajectory control. Compared to a solid microrobot, the lattice microrobot requires only one-sixth of the laser energy to achieve three times the motion speed and enables it to squeeze through confined spaces by adapting frictional interactions with its environment.

Despite these advances, several limitations remain to be addressed in future work. The use of purely optical actuation, while offering high spatial precision, limits robustness in dynamically changing or optically heterogeneous environments. Compared to optical tweezers, which are primarily suited for manipulating nanoscale objects and are highly sensitive to local heating and optical path quality, our system offers enhanced robustness for mesoscale object manipulation with reduced sensitivity to environmental optical heterogeneity. To overcome these constraints, future implementations may incorporate magnetic nanoparticles into the lattice structure, enabling large-scale locomotion through magnetic fields while retaining precise positioning via localized light stimulation. Additionally, the closed-loop control employs a waypoint sequence strategy, which is particularly suited to our microrobot's discrete, stepwise locomotion and periodic feedback constraints, as it allows precise navigation with minimal angular adjustments compared to path-following methods that require dense reference points. This approach leverages the microrobot's limited angular resolution and intermittent localization to optimize travel efficiency and robustness, as detailed in our analysis of locomotion characteristics, feedback timing,

and control precision. Future improvements in real-time localization and actuation smoothness could enable more advanced path-following control[57]. Future integration of real-time localization and continuous control algorithms could enhance trajectory tracking and responsiveness. It is also worth noting that although femtosecond lasers were used in this study, this choice was driven by equipment availability. In principle, the photothermal actuation of the hydrogel can be achieved using continuous-wave lasers, which may facilitate broader adoption of the technology in practical applications.

The structural and functional features of LSMR suggest potential in several domains. First, the low actuation energy requirement of the microlattice design allows a higher number of microrobots to be independently controlled under a single light source, especially when using beam-splitting techniques like spatial light modulators (SLM) or digital micromirror devices (DMD). This makes LSMR a promising candidate for high-precision multi-agent control. Second, due to its large specific surface area, the LSMR can carry a greater payload of functional agents compared to solid microrobots. This feature could be advantageous in applications such as localized drug delivery or biochemical sensing. For example, drugs could be released at target sites via localized heating, offering spatiotemporal control without requiring constant optical trapping.

## Methods
### Materials
N-isopropylacrylamide (NIPAM), N, N′-Methylenebisacrylamide (MBA), lithium phenyl-2,4,6-trimethylbenzoylphosphinate (LAP), poly-aminobenzene sulfonic acid functionalized single-walled carbon nanotubes (SWNT), ethylene glycol (EG), triethanolamine (TEA), Dextran.

Precursor configuration: 5 mg of SWNT was first dissolved in 1 mL of EG and ultrasonically shaken for 20 min to make it well dispersed. Then the supernatant was centrifuged at 7000 rpm (approximately 3380×g) for 25 min, and 500 μL of the supernatant was taken, to which the monomer NIPAM (450 mg), cross-linking agent MBA (50 mg), photoinitiator LAP (12 mg), and photosensitizer TEA (450 μL) were added sequentially. Finally, the sample was dispersed by ultrasonic waves for 20 min to make it fully dissolved and dispersed.

Sacrificial layer preparation: Firstly, the aqueous Dextran solution was prepared according to the ratio of 1:5 (w/w) and heated at 90 °C for 30 min until completely dissolved. Next, the glass substrate was ultrasonically cleaned with anhydrous ethanol, deionized water for 15 min, and dried. Next, the cleaned glass sheet was subjected to oxygen plasma treatment for 5 min to enhance the degree of affinity between the surface and water. Then, the aqueous Dextran solution configured above was spin-coated on the glass sheet at 8000 rpm (30 s). Finally, the glass sheets were baked on a heating table (180 °C) for 2 min.

### Laser direct writing and post-processing
A femtosecond laser beam (repetition frequency 80 MHz, pulse width 120 fs, center wavelength 780 nm, Discovery, Coherent Corp) was tightly focused through an oil-immersion objective (Olympus, ×40, NA 1.35) into a drop of precursor on glass. The structures were written directly with 400 nm layer spacing and 300 nm line spacing to microstructures assembled in Inventor. The laser power for direct writing was 25 mW, and the speed of direct writing was 6 mm/s. After laser exposure, the unpolymerized precursors were cleaned by immersion in propylene glycol methyl ether acetate for 20 min. Finally, the structures were rinsed with isopropanol (IPA) for 5 minutes to remove residual developer and then dried using a gentle stream of air.

### Actuation environment and optical setup
The prepared structures were adhered to a Dextran sacrificial layer. The glass substrate with the microstructures was placed under an

optical microscope (Olympus, IX83), and approximately 200 μL of deionized water was added dropwise to submerge the structures and dissolve the sacrificial layer, at which point the structures were freed from the substrate bond and rested on the glass substrate. Afterwards, a femtosecond laser beam (repetition frequency of 80 MHz, pulse width of 120 fs, center wavelength of 780 nm, Discovery, Coherent Corp) was focused through an air mirror (Olympus, 10×) and irradiated vertically on the microstructure through a vibrating mirror, a 2D piezoelectric carrier stage. The galvanometer controlled the beam to scan in the focusing plane, and the displacement stage drove the glass slice to adjust the position of the microstructure.

## SEM and mechanical characterization

The SEM images were obtained from a field-emission SEM (Nova NanoSEM 450) with an acceleration voltage of 10 kV. An in situ micromechanical test system (FemtoTools, FT-MTA02) for stiffness testing of material blocks. An FT-100000 microforce transducer with a range of ±100000 μN and a resolution of 5 μN was used, with a square tip size of $50 \times 50$ μm.

## Closed-loop control based on computer vision

The control process is divided into three key steps: position evaluation, orientation adjustment, and forward movement. First, the system assesses whether the robot has reached the target by comparing the detected geometric center of the robot with the target position. If the target is not reached, the system proceeds to orientation adjustment and forward movement in sequence. The current loop ends as soon as either step is completed, and the next loop begins using updated data for detection and control. A CCD continuously captures the motion of the light-driven peristaltic robot. The captured images are preprocessed using the OpenCV library (an open-source computer vision toolkit) to detect the robot's orientation and spatial position. The relative angle and distance between the robot's geometric center and the target are then calculated. To minimize unnecessary rotations, the smallest possible angle is always selected. These results are mapped to the galvanometer's kinematic coordinate system, generating control data to guide the laser spot for precise movement.

## Data availability

All data needed to evaluate the conclusions in the paper are present in the manuscript and Supplementary Information. The data are also available upon request from the corresponding author. Source data are provided with this paper.

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

## Acknowledgements

This research was financially supported by the National Key Research and Development Program of China (grant number 2021YFF0502700), the National Natural Science Foundation of China (grant numbers 52275429 and 62205117), an Innovation Research Project of Optics Valley Laboratory (grant number OVL2021ZD002) Hubei Provincial Natural Science Foundation of China (No. 2022CFB792), Young Elite Scientists Sponsorship Program by CAST (No. 2022QNRC001), Fundamental Research Funds for the Central Universities (No. YCJJ20242405), West Light Foundation of the Chinese Academy of Sciences (No. xbzgzdsys-202206) and Knowledge Innovation Program of Wuhan-Shuguang. We thank Xinger Wang and Mingjing Li from Huazhong University of Science and Technology for suggestions on the visualization of this paper. We thank the Analytical and Testing Center of Huazhong University of Science and Technology, the Center of Optoelectronic Micro&Nano Fabrication and Characterizing Facility, Wuhan National Laboratory for Optoelectronics of Huazhong University of Science and Technology for the support in device fabrication.

## Author contributions

M.Z. and Y.L. contributed equally to this work. M.Z. and W.X. conceived the ideas and designed the study. M.Z. designed the microstructure, developed the preparation process, performed the experiments, and analyzed the data. Y.L. designed and wrote the code program for the closed-loop drive and participated in the experiment of the closed-loop drive. C.D. provided the precursor configuration methodology. X.F. completed the mechanical characterization and photoresponsive performance testing. F.C. and S.S. constructed and maintained the laser direct-write and closed-loop feedback control system. Z.Z., H.H., S.X., L.D., L.L., T.S., and H.G. discussed the results. All authors wrote the manuscript together.

## Competing interests

The authors declare no competing interests.
