## [Transparent Peer Review file · Nature Communications]

Light-driven Lattice Soft Microrobot with Multimodal Locomotion

Corresponding Author: Professor Wei Xiong

Version 0:

Reviewer comments:

Reviewer #1

(Remarks to the Author)

The paper presents a light-driven lattice soft microrobot (LSMR) capable of peristalsis, in situ rotation, and hopping locomotion, demonstrating the advantage of lattice structures in enhancing energy conversion efficiency and multimodal motion. Below are my detailed comments and suggestions for improvement:

1. Line 41-42: The authors should clarify what "these limitations" refer to. Are they referring to replicating the movement patterns of natural organisms? If so, why is this important for practical applications? Instead of simply mimicking biological systems, why can't we engineer more efficient locomotion patterns? What "practical applications" and "field" do they refer to? Please be more specific about your claim.
2. Line 49-51: The "bend-extend mechanism" is mentioned as a limitation. Please clarify this mechanism further. If the primary drawback is the inability to mimic versatile motions but it still achieves fast movement, why is this limitation significant?
3. Line 63: The authors claim the fastest peristaltic movement speed. Why does it matter that the locomotion is specifically peristaltic? A more useful comparison would be to benchmark the LSMR against the fastest locomotion speeds for soft robots of a similar scale, regardless of the locomotion mode. The author presented the comparison in Fig. 6h and the robot presented is not that fast compared to others.
4. Line 70: Please clarify what "longstanding challenges" the authors are referring to. Are these related to locomotion efficiency, energy conversion, or something else?
5. Line 71-73: The statement about "wide-ranging implications in biomedicine, bionics, and micromechanical engineering" is too vague. Please specify potential applications and how the LSMR could address current challenges in these fields.
6. Line 121: Both the diameter and length of the lattice edges are varied in the experiments. Would it be possible to fix one parameter while varying the other to isolate the effects on deformation and performance?
7. Please label all the structures in Supplementary Movie 1.
8. Do not speed up Supplementary Movie 5. The fast pace makes it difficult to observe how the LSMR navigates through the narrow slit. A slowed-down version would be more informative.
9. The authors need to provide more details on how the stiffness measurements of the lattice blocks shown in Figure 2f and Supplementary were conducted. What was the experimental setup used to obtain these results?
10. Line 163: Please elaborate more on the rationale for selecting L4D4 as the unit structure for the LSMR.
11. Line 207: What are the dimensions and weight of the solid structure used in comparison with the lattice structure? Are both structures equivalent in weight and 2D dimensions?
12. Both manual and closed-loop control systems show significant trajectory-following errors. Can the authors explain why these errors occur and suggest ways to minimize them?
13. What is the maximum hopping height achieved by the LSMR? Could the hopping motion be utilized to overcome obstacles of a certain height?
14. The discussion on future work is brief. The authors should provide more detailed elaboration on potential real-world applications of the LSMR and how they plan to address the current limitations to make the robot practical for such use cases.

Reviewer #2

(Remarks to the Author)

The authors introduce a light-actuated soft microrobot that leverages a lattice structure. I think this work is novel, complete,

and useful to the community, and will be ready for publication after addressing a few comments:

Firstly, I think the introduction is very general, and could benefit from a more in-depth discussion of the work that immediately preceded this. Have lattices been used before to show locomotion? Has other work focused on the compression of lattices? Have other works used this principle effectively in other robots at any scale?

Additionally, the language in the introduction is self-important—phrases such as “...exhibits, by far, the fastest...”, with qualifiers like “unmatched,” “superior,” “remarkable,” etc., should be evident from the work, not your language.

Figure 6h should have a counterpart in the supplementary materials that plots speed in some absolute unit (like mm/s or m/s) vs body length.

Are there constraints to using light-based methods? I want a more in-depth discussion of possible applications that could use this robot and system, as is and in the future. How will you apply light inside the human body? Is this done already, and how? What are the “applications in medicine, engineering, and other fields,” and why is your solution the best, or even a plausible solution? Discussing and answering these important motivational questions would strengthen your work.

The paper, overall, does not read as a story, as much of the text starts with “In Figure X, [...]” Instead of writing subplot by subplot, I recommend stating your result, then citing the figures in parentheses. For example, instead of “Fig. 2e shows the response time and shrinkage ratio of both a lattice structure (L6D6) and a solid structure under light stimulation. The lattice structure achieved 30.58% (95,176 μm^3) volume shrinkage in just 1.46 seconds, whereas the solid structure took 2.50 seconds to shrink by only 10.55% (22,788 μm^3).”, say “The lattice structure achieved 30.58% (95,176 μm^3) volume shrinkage in just 1.46 seconds, whereas the solid structure took 2.50 seconds to shrink by only 10.55% (22,788 μm^3). (Fig. 2e)”.

Additionally, instead of saying, “[result], as illustrated in Fig. X.”, say “[result] (Fig. X).” This will prevent distraction. There are many instances of both of these, but I think the manuscripts readability will improve drastically with these changes.

Small things: the two parts of Fig 3a could likely be combined (indicating the laser location on the top-down view).

Supplementary movies are referenced incorrectly—in supplementary materials document it says there are 10 videos, and in the text, it references video 7 when it meant video 5. Just check these over and make sure they are consistent.

Reviewer #3

(Remarks to the Author)

In this article the authors proposed to propel 100 μm microbot with light using two different kinds of motions which are peristaltic motion and a combination of marangoni and buoyancy induced flow.

The novelty of this work is to largely increase the efficiency by using a 3D printed metamaterial approach as well as a nanotube functionalisation which increase the deflection of static part as previously reported by some of the authors in reference 31.

The article is well organised and written and the results are clearly presented. They demonstrate the capacity and original contribution of the author approaches to mobile microrobot. The approach proves to be faster (but less than a factor 2) than other light induced peristaltic microrobot report in reference 9 and demonstrate the ability to switch between two motion type.

Before considering publication, I think the following question/comments should be addressed :

- I think the potential application of such microrobot should be illustrated more precisely. This is crucial to compare its advantage to other technique. For example, in which scenario would it be better to use the authors proposed system than microrobot based on optical tweezers which have the same environment constraints (i.e. environment should allow light to be focus on the microrobot) ?

- I don't understand the point of the speed model presented in equation 2. Indeed the best parameters for the robot actuation seems to be set afterwards directly using an empirical approach. The goal of this model should be stated clearly. It should be compared to experiment data to be validated.

- How the numerical different coefficients of equation 2 are found (A1,A2,xo,dx,yo,A,frac,xO1,xO2,K1,K2)? Is it from an experimental fit ?

- I think the energy efficiency as it is defined currently in the article is not a pertinent criteria. Indeed, at this scale, the kinetic energy is negligible compared to the drag friction.

To illustrate this let assumes that the robot is a 100 μm sphere, of density 1 travelling at 10 $\mu\text{m}/\text{s}$ in water which are conditions close to the authors robot.

In this case, assuming a Stokes law for the drag (i.e. the friction due to fluid viscosity) we obtain :

$$E_{\text{kinetic}} = 10^{-22} \text{ Joule}$$

$$F_{\text{drag}} = 10^{-11} \text{ Newton}$$

Therefore the distance to dissipate the kinetic energy with the drag is only 10 nm. This means that an estimation of the duration when the energy of the laser is converted into kinetic energy is only 10 nanometer. After this, all the laser power transferred to the robot is dissipated in drag.

To conclude, it would make more sense to consider a power ratio (Drag force on the robot/ power of the laser) or simply the speed of the robot.

- The rotation ability of the robot could be better presented to understand its angular precision. Figure 4 f display the rotation with time but there is no information on the angular precision that the robot can reach. Could this somehow be tuned by the laser power (or other parameters) ? I think this is a crucial parameter to assess the precision the microrobot could perform in path following.

- Could you provide details on the method to adjust direction in the closed loop control scheme as display in Fig S8 ?

- In general the closed loop is very crude and could inspire from other closed loop in the microrobotics literature . More detail on how to improve it should be considered. Please find in the following an article illustrating few methods : Dahroug, Bassem, et al. "Some examples of path following in microrobotics." 2018 International Conference on Manipulation, Automation and Robotics at Small Scales (MARSS). IEEE, 2018.

- Does a femto second laser is required to actuate the robot (as in this presented work) ? Would it work with a continuous laser ? I think this is an important information to assess the complexity of powering such microrobot.

Minor Comments

The authors state ; "Manual annotation of target waypoints ensures accurate navigation."
Could you elaborate on what this sentence mean ?

Version 1:

Reviewer comments:

Reviewer #1

(Remarks to the Author)

The authors have carefully addressed all my comments and concerns. I appreciate the efforts from the authors for the careful revision.

Reviewer #2

(Remarks to the Author)

Thank you for thoroughly addressing my concerns!

Reviewer #3

(Remarks to the Author)

The authors have addressed well most of the points I previously mentioned and the overall quality of the manuscript is greatly improved. I still have one major point and three minor points before considering publications.

Major :

on comment 3

It is still unclear how the 4 curves of figure S6a are used to fit all the following parameters(A_1 , A_2 , x_0 , dx , y_0 , A , frac , x_{01} , x_{02} , K_1 , K_2) in Equation 2.

The fitting methods of each parameters need to be explained. This is crucial to achieve the link between the material property on the robot speed modulation. Without it the model seems useless to others researcher.

Minor :

Comment 4

I appreciate the effort of the authors to try to measure force apply by their robot which is truly a great challenge at this scale. Nonetheless I still think that comparing the speed (or the speed in body length per seconde) is more relevant than an energy efficiency based on kinetic energy. I would therefore suppress the line mentioning kinetic energy.

Comment 6,

Thank you for the detail on the closed loop, I think these should be summarised in the article or supplementary material.

Comment 7/

Thank you for putting in perspective your closed loop with existing one in the literature.

I think the explanation for choosing waypoint sequence is valid. The article would greatly benefit a sentence or two to justify the explanation of the waypoint method compared to others in this particular case.

Version 2:

Reviewer comments:

Reviewer #3

(Remarks to the Author)

The Authors have answer all my remaining questions. The article is ready for publication.

**Manuscript Revision Details**

Journal: Nature Communications

Ms. No.: NCOMMS-24-79506

Ms. Title: Light-driven Lattice Soft Microrobot with Multimodal Locomotion

Authors: Mingduo Zhang^{1†}, Yuncheng Liu^{1†}, Chunsan Deng¹, Xuhao Fan¹, Zexu
Zhang¹, Shaoxi Shi¹, Fayu Chen¹, Huace Hu¹, Songyan Xue¹, Leimin Deng^{1,2}, Lige Liu³,
Tao Sun³, Hui Gao^{1,2}, Wei Xiong^{1,2*}

Dear editors and reviewers,

The authors would like to thank the editor and all reviewers for their constructive
comments and suggestions for our manuscript. We are glad to get all your positive
assessments of our submission and appreciate for giving us the opportunity to submit a
revised draft. We have incorporated the suggestions made by the reviewers and revised
the manuscript accordingly.

Below, we provide a point-by-point response (in blue) to the reviewer's comments (in
black) and point out the places where we have revised. The modifications have also
been highlighted in red color in the revised manuscript.

Thanks again for your time in reviewing our manuscript. Looking forward to hearing
from you soon.

Yours sincerely,

Wei Xiong (on behalf of all authors)

weixiong@hust.edu.cn

**Revision Summary**

**Manuscript:**

- 1. Page 1, Line 23, removed phrases of primacy, exaggerated and subjective language.
- 2. Page 2, Line 33, revised vague language in Introduction and refined the logic of the
writing.
- 3. Page 2, Line 49, added description of research on lattice-structured hydrogel
materials.
- 4. Page 3, Line 53, revised review of research on deformation strategies for microrobots.
- 5. Page 3, Line 65, modified primacy, exaggerated and subjective, vague language.
- 6. Page 4, Line 85, modified language to improve readability.
- 7. Page 5, Line 119, added description of data sources for mathematical modeling.
- 8. Page 6, Line 124, modified language to improve readability.
- 9. Page 6, Line 142, modified language to improve readability.
- 10. Page 6, Line 151, added description of stiffness test methods.
- 11. Page 7, Line 169, added justification for the choice of microrobot parameters.
- 12. Page 8, Line 183, modified language to improve readability.
- 13. Page 8, Line 186, modified language to match the new Fig. 3a.
- 14. Page 8, Line 195, added explanation of the source of the fitted parameters in the
mathematical model.
- 15. Page 9, Line 202, updated Supplementary Figure number.
- 16. Page 9, Line 204, updated Supplementary Figure number.
- 17. Page 9, Line 222, added explanation that solid microrobots are the same size as
lattice microrobots.
- 18. Page 9, Line 226, modified language to improve readability.
- 19. Page 10, Line 235, modified view of the schematic of Fig. 3a.
- 20. Page 11, Line 247, modified language to improve readability.
- 21. Page 11, Line 258, updated Supplementary Movie number.
- 22. Page 11, Line 258, modified language to improve readability.
- 23. Page 11, Line 266, modified language to improve readability.
- 24. Page 12, Line 278, added research of the angular accuracy of the in-situ rotation
- 25. Page 12, Line 292, updated Supplementary Movie number.
- 26. Page 12, Line 301, updated Supplementary Figure number.
- 27. Page 13, Line 316, modified language to improve readability.
- 28. Page 14, Line 321, updated Supplementary Figure number.
- 29. Page 14, Line 322, updated Supplementary Movie number.
- 30. Page 14, Line 329, updated Supplementary Movie number.
- 31. Page 14, Line 336, modified vague language to add explanation of using redundant

- waypoints to improve control accuracy.
- 32. Page 16, Line 354, modified language to improve readability, updated
Supplementary Figure number and updated Supplementary Movie number.
- 33. Page 16, Line 368, modified language to improve readability, updated
Supplementary Figure number and updated Supplementary Movie number.
- 34. Page 16, Line 379, updated Supplementary Figure number.
- 35. Page 17, Line 390, modified language to improve readability.
- 36. Page 17, Line 393, modified language to improve readability, removed phrases of
primacy, exaggerated and subjective language.
- 37. Page 18, Line 400, in the second paragraph of the Discussion, a discussion of the
limitations of the present work is added, a comparison with optical tweezers-driven
microrobotics is added, and a discussion of possible ideas for closed-loop feedback
control, using continuous lasers, to improve the present work is added.
- 38. Page 18, Line 415, added discussion of future applications of lattice microrobots.

**Supplementary information:**

- 1. Page 1, Line 14, updated Supplementary Figure number.
- 2. Page 1, Line 18, updated Supplementary Movie number.
- 3. Page 3, Line 54, added explanation of the source of the fitted data.
- 4. Page 7, Line 105, added the stiffness test setup as well as experimental images and
diagrams of the test procedure.
- 5. Page 9, Line 137, added data plots of in-situ rotation angle accuracy driven by
different laser powers.
- 6. Page 11, Line 152, added a summary plot of the motion speed of a soft micro-robot
using μ m/s as the coordinate system.

**Response to reviewer comments**

**Reviewer: 1**

The paper presents a light-driven lattice soft microrobot (LSMR) capable of peristalsis,
in situ rotation, and hopping locomotion, demonstrating the advantage of lattice
structures in enhancing energy conversion efficiency and multimodal motion. Below
are my detailed comments and suggestions for improvement:

**Response:**

We sincerely appreciate your time and effort in reviewing our manuscript. Your
insightful comments and suggestions are invaluable in improving the quality of our
work. Below, we provide detailed responses to each of your comments.

The Reviewer #1's Comment 1

Line 41-42: The authors should clarify what “these limitations” refer to. Are they referring to replicating the movement patterns of natural organisms? If so, why is this important for practical applications? Instead of simply mimicking biological systems, why can’t we engineer more efficient locomotion patterns? What “practical applications” and “field” do they refer to? Please be more specific about your claim.

Response:

We recognize that the description in the original manuscript is vague. In the revised manuscript we have replaced the original vague description with a specific description. To enhance clarity and accurately convey our intended message, we have revised the manuscript as follows: 1. Clearly defining the primary challenge in this research field: achieving controlled, efficient, and adaptive motion in small-scale systems. 2. De-emphasizing the difficulties of mimicking natural microorganisms and instead focusing on how bio-inspired approaches have already led to the development of various microrobot systems. 3. Provide detailed references to “limitations”. Below, we offer detailed explanations for the changes we made.

The limitations we aimed to describe are those outlined in our original manuscript: *“Despite this interest, significant challenges remain in achieving efficient, controllable locomotion at such a small scale.”* rather than the mere replication of natural organisms’ movement patterns. Even drawing inspiration from the locomotion of microorganisms, our objective is to develop more efficient and controllable motion strategies to enhance the overall capabilities of microrobots. In this respect, we align with the reviewer’s perspective.

In response to the question, “Instead of simply mimicking biological systems, why can’t we engineer more efficient locomotion patterns?”. We sincerely apologize for misunderstanding caused by our unclear wording. In fact, studying biological locomotion provides significant advantages in improving certain aspects of microrobot performance. Moreover, in this study, we have demonstrated how engineering can enhance locomotion by replacing solid structures with a lattice-based design, which is not found in natural microorganisms. This approach exemplifies how engineered solutions can surpass biological constraints to achieve better performance. Our main point is that natural microorganisms exhibit greater flexibility in movement and superior environmental adaptability compared to current artificially fabricated soft microrobots. These characteristics, enhanced locomotion flexibility and adaptability, are essential for bringing soft microrobots closer to real-world applications. This is the core message we intended to convey.

We appreciate the reviewer pointing out the ambiguity in our original description of practical applications. In response, we have revised the manuscript to more clearly

articulate the envisioned application scenarios that our microrobot design could
potentially benefit. Although our current study does not include experimental
demonstrations of real-world applications, the proposed strategy enables the
development of hydrogel-based soft microrobots with enhanced response performance,
the ability to navigate through narrow gaps, and precise closed-loop motion control via
computer vision. These are the core contributions of our work.

Due to space constraints in the Introduction section, we refrained from elaborating on
specific application domains there. Instead, a more detailed discussion is provided in
the Discussion section of the revised manuscript. Two representative application areas
are highlighted: High-Precision Multi-Agent Control: The reduced power demand of
the LSMR allows for wide-field light actuation using DMD/SLM-based optical systems,
enabling parallel manipulation of multiple microrobots. High Drug-Loading Capacity
for Controlled Release: The increased surface area afforded by the LSMR's lattice
structure enhances drug adsorption capacity. In order to avoid repetition, we have
provided an outlook on future application scenarios of LSMR in our response to
comment 5 and comment 14, and we have deleted the introduction on practical
applications and domains here.

These examples, while not directly demonstrated in the current study, help to illustrate
the broader significance and translational potential of our approach. We hope this
clarification helps better convey the relevance of our work to the field of soft
microrobotics.

**Revisions:**

On page 2, lines 33-44 of the Revised Manuscript.

We modified the first paragraph of the introduction to be more specific and explicit as
“Microscale robotics has gained significant attention due to its potential applications in
biomedicine, environmental monitoring, and precision micromechanics.¹⁻³ Among
them, submillimeter soft microrobots are particularly promising due to their small size,
adaptability, and diverse actuation possibilities.⁴⁻⁸ Biological microorganisms have
evolved highly efficient and adaptable movement strategies to navigate complex
environments.^{9,10} Inspired by these mechanisms, bioinspired motion patterns such as
peristalsis^{11,12}, inchworm-like crawling^{13,14}, walking¹⁵⁻¹⁷, and swimming^{18,19} have been
applied to the design of soft microrobots, expanding the range of achievable locomotion
strategies.²⁰⁻²² However, achieving controlled, efficient, and adaptive motion in such
small systems remains a considerable challenge. The constraints in actuation efficiency,
and limited degrees of freedom continue to hinder the realization of highly versatile and
multimodal locomotion. The integration of advanced actuation materials, precise
actuation strategies and bioinspired structural designs offers a potential solution to
improve the locomotion efficiency, adaptability.”

The Reviewer #1's Comment 2

Line 49-51: The "bend-extend mechanism" is mentioned as a limitation. Please clarify this mechanism further. If the primary drawback is the inability to mimic versatile motions but it still achieves fast movement, why is this limitation significant?

Response:

We thank the reviewer for this insightful comment. In the revised manuscript, we have clarified the meaning of the “bend-extension mechanism.” We have also explained more clearly why its limited adaptability is a significant constraint in some situations. Additionally, we emphasized that our proposed approach uses the high spatial resolution of light stimulation. This allows for localized deformations throughout a fully soft microrobot. As a result, it enables programmable and multimodal motion. This distinction highlights the advantage of adaptability in diverse environments while maintaining a balanced and objective discussion of the existing strategies. Below, we provide further explanation.

Rather than positioning the “bend-extension mechanism” as a limitation, our primary aim is to highlight the distinctive characteristics of our proposed actuation strategy. The “bend-extension mechanism” has demonstrated relatively rapid locomotion in controlled environments, as acknowledged in our revision. However, such systems generally rely on fixed deformation pathways, and their motions are typically governed by global, predefined actuation patterns. This restricts their ability to perform context-dependent or reconfigurable tasks, which limits their effectiveness in complex environments where real-time adaptability is essential—for instance, when navigating through confined, irregularly shaped channels or interacting with heterogeneous materials.

In contrast, our proposed actuation strategy makes use of light's high spatial precision to induce localized deformations across the structure of a fully soft microrobot. This enables real-time control over shape and function, supporting programmable and multimodal locomotion. We believe this capability for adaptable and diversified movement, while maintaining comparable response speed, represents a significant advancement over conventional bend-extension-based strategies.

We hope this clarification adequately addresses the reviewer's concern.

Revisions:

On page 2, lines 53-64 of the Revised Manuscript.

We modified the introduction to be more specific and explicit as “**Meanwhile, structural design plays a crucial role in shaping the motion strategies of soft microrobots. Many existing approaches rely on predefined periodic deformations to achieve movement, demonstrating relatively rapid locomotion in controlled environments.**”^{13-15,36} However,

these methods often encounter challenges in complex surroundings, where the ability
to reconfigure and adapt to external stimuli is essential for effective navigation. One
approach to address this challenge is to enable localized deformations throughout a
fully soft microrobot, allowing different regions of the body to undergo shape changes
in a programmable manner without being constrained to a single predefined
deformation mode. Leveraging the high spatial resolution of light stimulation, precise
control over localized shape transitions can be achieved, supporting multimodal motion.
By integrating a fully soft body structure with precision light actuation, this design
improves adaptability in complex terrains and enhances control over deformation and
movement, enabling microrobots to operate more effectively in diverse environments.”

The Reviewer #1's Comment 3

Line 63: The authors claim the fastest peristaltic movement speed. Why does it matter that the locomotion is specifically peristaltic? A more useful comparison would be to benchmark the LSMR against the fastest locomotion speeds for soft robots of a similar scale, regardless of the locomotion mode. The author presented the comparison in Fig. 6h and the robot presented is not that fast compared to others.

Response:

We thank the reviewer for the valuable comment and the opportunity to clarify our intent. We recognize that our original emphasis on achieving the “fastest peristaltic movement speed” may have led to a misunderstanding regarding the primary goal of this study. Our objective is not to pursue the absolute highest locomotion speed among soft microrobots, but rather to explore how the integration of advanced actuation materials, precise actuation strategies, and bioinspired structural designs can collectively enhance the efficiency, adaptability, and functional applicability of soft microrobots. The relatively fast peristaltic speed observed in our LSMR is a result of these combined strategies, not the central focus of our research.

Considering the reviewer’s and editor’s suggestions, we have removed the phrase “the fastest” from the revised manuscript to avoid misdirecting the reader. Furthermore, in Fig. 6h, we have limited our comparison to soft microrobots that rely on body deformation for locomotion—an approach consistent with our own. This provides a fair and focused benchmarking context aligned with the specific domain addressed in our study.

Below, we provide a more detailed explanation of the significance of peristaltic motion and the rationale behind our comparison framework.

We recognize the importance of clarifying the significance of peristaltic motion. Peristalsis is widely observed in organisms like earthworms and *Euglena*, which use this mode of propulsion for traversing narrow spaces and uneven surfaces. By mimicking this biological mechanism, we ensure that our robot can operate efficiently in complex terrains. Unlike rigid mechanisms, peristaltic robots can deform continuously along their body length, providing superior flexibility when operating in confined or irregular environments. This makes them particularly suitable for applications such as minimally invasive surgery or pipeline inspection. The control of peristaltic motion relies on generating a wave-like contraction/extension pattern, which simplifies actuation compared to systems requiring multiple independent motors. Additionally, it reduces energy consumption while maintaining robust forward progress. A key demonstration of this advantage in our study is the LSMR’s ability to squeeze through constrictions as narrow as 75% of its resting width under laser actuation. This capability highlights how peristaltic motion, combined with a lattice-enhanced soft

structure, allows for high adaptability in environments where rigid or non-deformable
robots would fail to reach the target. For example, Xiong et al. reported a light-
controlled soft biomicrobots (called “Ebot”) based on *Euglena gracilis* that are
capable of performing multiple tasks in narrow microenvironments including intestinal
mucosa with high controllability, deformability and adaptability.¹ It is based on
peristalsis and body deformation that the Ebot achieves movement in confined
environments.

We appreciate Reviewer#1’s suggestion that benchmark the LSMR against the fastest
locomotion speeds for soft robots of a similar scale, regardless of the locomotion mode.
And we have revised the manuscript to clarify our research focus. The primary goal of
our study is not to develop the fastest microrobot within this scale range but rather to
explore how the integration of advanced actuation materials, precise actuation
strategies, and bioinspired structural designs can improve the efficiency, adaptability,
and functional applicability of soft microrobots. The relatively high peristaltic
locomotion speed achieved in our work is a result of the proposed lattice-enhanced
actuation strategy.

Locomotion speed in soft microrobots is influenced by multiple factors, including:

**Material response speed:** Actuation dynamics vary significantly across materials. For
example, metallic and elastomeric actuators typically exhibit faster response times than
hydrogel-based actuators due to differences in thermal and mechanical properties.

**Locomotion strategy:** Jumping and walking tend to achieve higher speeds compared
to peristalsis or crawling, as they rely on discrete, high-energy propulsion events rather
than continuous deformation.

**Actuation efficiency:** The ability to convert external energy into mechanical work
depends on multiple factors, including heat dissipation, material hysteresis, and energy
storage mechanisms.

**Environmental interactions:** The substrate properties, frictional conditions, and
surrounding medium significantly influence the achievable speed. Locomotion
strategies optimized for one type of environment may not perform equivalently in
another.

To provide a broader context for our work, we have incorporated a direct comparison
of locomotion speeds of soft microrobots with different actuation mechanisms in Fig.
6h, allowing for a more comprehensive benchmarking.

We acknowledge that some soft microrobots utilizing alternative locomotion strategies
achieve higher absolute speeds. However, as discussed above, locomotion speed alone
does not fully capture the advantages of peristaltic motion or the significance of our
structural design approach. The LSMR’s ability to combine multimodal locomotion
(including peristalsis, in situ rotation, and hopping) with structural flexibility
differentiates it from systems optimized purely for speed. Furthermore, its ability to

squeeze through constrictions demonstrates its environmental adaptability, which is a
key design consideration for soft microrobots intended for confined or structured
environments.

To further clarify our research focus, we have refined the manuscript to emphasize the
role of lattice-based structural design in improving microrobot performance, rather than
locomotion speed as the primary evaluation metric.

We appreciate the reviewer's valuable feedback, which has helped us refine our
discussion and improve the clarity of our manuscript. We hope these revisions
sufficiently address the concerns raised.

**References:**

1. Xiong, J. et al. Light-controlled soft bio-microrobot. *Light: Sci. Appl.* 13, 55 (2024).

**Revisions:**

On page 3, lines 65-77 of the Revised Manuscript.

We modified the last paragraph of the introduction to be more specific and explicit as
“To address the limitations above, we present a light-driven lattice soft microrobot
(LSMR) constructed from poly(N-isopropylacrylamide)-single-walled carbon
nanotubes (PNIPAM-SWNT) hydrogels with a truncated octahedron lattice structure to
enhance deformability and actuation efficiency. The lattice design reduces relative
density, allowing for greater flexibility and faster deformation under light stimulation
compared to solid microrobots made from the same hydrogel precursor material. The
LSMR generates localized body deformations under high-precision laser stimulation,
enabling fine control over shape transformation. By adjusting scanning frequency,
trajectory, and power, these deformations generate different locomotion modes,
including linear peristalsis, in situ rotation, and hopping. The LSMR exhibits a
peristaltic locomotion speed of 15.15 $\mu\text{m/s}$ (0.14 body lengths per second) and an in
situ rotation speed of 29.38°/s, alongside an energy conversion efficiency 16.49 times
higher than that of solid counterparts. The LSMR can squeeze through constrictions as
narrow as 75% of its resting width under laser actuation and autonomously follows
programmed paths with closed-loop feedback. These advancements illustrate that the
integration of lattice-based structural design and precise light-driven actuation
contributes to improved energy conversion efficiency, facilitates multimodal
locomotion, and enhances adaptability in constrained environments, which are critical
challenges in the development of soft microrobots, and may support future applications
in areas such as high-precision multi-agent control, high drug-loading capacity for
controlled release.”

The Reviewer #1's Comment 4

Line 70: Please clarify what "longstanding challenges" the authors are referring to. Are these related to locomotion efficiency, energy conversion, or something else?

Response:

We apologize for not clearly expressing this point in our original manuscript and appreciate the opportunity to further elaborate on the "longstanding challenges" mentioned. Below, we provide a detailed response to specify the key challenges our work addresses.

The phrase "longstanding challenges" in our original manuscript primarily refers to three interrelated difficulties in the field of light-driven soft microrobots:

Actuation efficiency: Many soft microrobots suffer from low energy conversion efficiency, especially hydrogel-based actuators, which exhibit slow response times and limited force output due to the material's inherent swelling and deswelling characteristics.

Locomotion adaptability: Traditional light-driven microrobots often rely on pre-programmed or structurally constrained movement patterns, limiting their ability to adapt to environmental changes and perform diverse locomotion modes.

Navigation in confined spaces: Achieving controlled movement in highly constrained environments, such as squeezing through narrow gaps or navigating irregular terrains, remains a critical challenge due to the difficulty of achieving precise, localized actuation.

Our study specifically targets these issues by integrating a lattice-based structural design with precise light-driven actuation. This approach enhances energy conversion efficiency, enables multimodal locomotion, and improves adaptability in confined spaces, as demonstrated in our experimental results.

Revisions:

On page 3, lines 77-81 of the Revised Manuscript.

We modified the last paragraph of the introduction to be more specific and explicit as “These advancements illustrate that the integration of lattice-based structural design and precise light-driven actuation contributes to improved energy conversion efficiency, facilitates multimodal locomotion, and enhances adaptability in constrained environments, which are critical challenges in the development of soft microrobots, and may support future applications in areas such as high-precision multi-agent control, high drug-loading capacity for controlled release.”

The Reviewer #1's Comment 5

Line 71-73: The statement about "wide-ranging implications in biomedicine, bionics, and micromechanical engineering" is too vague. Please specify potential applications and how the LSMR could address current challenges in these fields.

Response:

We sincerely thank the reviewer for pointing out the vagueness in our original description of potential applications. Although the current manuscript does not include experimental demonstrations of real-world applications, we believe that, based on the experimental results and the inherent advantages of the lattice-based structure, our LSMR system holds promise for future use in areas such as high-precision multi-agent control and high drug-loading capacity for controlled release.

In the revised manuscript, we have removed the previously vague expression of "wide-ranging implications" and replaced it with these two specific application directions. Due to space limitations, a more detailed discussion of the potential applications is now included in the Discussion section. A summary of these envisioned directions is provided below:

In the field of soft microrobotics, achieving high-precision and independent control of multiple robots is a significant challenge. Among various control methods (such as magnetic, pH-based controls), optical control stands out due to its high spatial resolution, enabling independent operation of multiple soft microrobots. However, when controlling multiple robots simultaneously, traditional optical control methods typically use devices like Digital Micromirror Devices (DMD) or Spatial Light Modulators (SLM) to split laser beams.^{1,2} This process reduces the energy of each beam, which limits the number of independently controlled units under a fixed total laser power. Our LSMR requires less energy compared to conventional solid microrobots due to its unique lattice structure. Consequently, it allows for the independent control of a larger number of units under the same light source conditions, thereby significantly increasing the upper limit of multi-body control.

Furthermore, the lattice structure of LSMR offers another important advantage—its higher specific surface area compared to solid microrobots. This structural feature enables LSMR to carry more functional substances, such as drug molecules, and achieve controlled release under specific conditions.^{3,4} This capability has significant implications for targeted therapy and controlled drug delivery. For instance, in cancer treatment, LSMR can be guided to the lesion site using external optical signals and triggered to release drugs locally through heating or other stimuli. Moreover, in tissue repair or regenerative medicine, LSMR can serve as a carrier for growth factors or stem cells to promote wound healing and new tissue formation.

Additionally, the design flexibility of LSMR allows it to easily integrate with other
technologies, such as magnetic control or chemically responsive materials, enabling
multifunctional capabilities. Combining optical control with magnetic control can
enhance the navigation ability of robots and improve their adaptability to complex
environments. Similarly, incorporating pH-responsive materials into LSMR can
provide additional environmental sensing capabilities in dynamic settings.

Overall, the distinctive characteristics of LSMR and its potential applications
demonstrate its broad prospects in the fields of biomedicine, bionics, and
micromechanical engineering. We hope to explore more possibilities through further
research and development, contributing to the advancement of these technologies.

While these applications are not yet experimentally validated, they represent promising
directions for future development. We believe that integrating our approach with
additional control modalities (e.g., hybrid optical-magnetic actuation or fiber-optic
delivery systems) may enable their realization.

We hope this revision and clarification address the reviewer's concern and improve the
clarity of our manuscript regarding the potential relevance and future utility of the
proposed design.

**References:**

- 1. Xin, C. et al. Light-triggered multi-joint microactuator fabricated by two-in-one
femtosecond laser writing. *Nat. Commun.* 14, 4273 (2023).
- 2. Kim, M., Yu, A., Kim, D., Nelson, B. J. & Ahn, S. Multi-agent control of laser-
guided shape-memory alloy microrobots. *Adv. Funct. Mater.* 33, 2304937 (2023).
- 3. Chen, M., Lu, J., Hou, J. & Zhao, Y. Fabrication of hybrid rod-like mesoporous silica
nanocarriers for responsive drug release and combined photo-chemotherapy. *Colloids*
*Surf., A* 648, 129227 (2022).
- 4. Cai, L., Wen, Y., Lin, Z., Qian, H. & Deng, X. Synthesis of magnetic thermo-
sensitive polymeric microspheres for controlled release via magnetic induction. *Acta*
*Polym. Sin.* 012, 846–851 (2012).

**Revisions:**

On page 3, lines 80-81 of the Revised Manuscript.

We modified the last paragraph of the introduction to be more specific and explicit as
“....., and may support future applications in areas such as high-precision multi-agent
control, high drug-loading capacity for controlled release.”

The Reviewer #1's Comment 6

Line 121: Both the diameter and length of the lattice edges are varied in the experiments. Would it be possible to fix one parameter while varying the other to isolate the effects on deformation and performance?

Response:

We sincerely thank the reviewer for the insightful suggestion. While we attempted to reorganize our experimental design by fixing one structural parameter (rod length or diameter) and varying the other, we ultimately retained our original parameter matrix in Fig. 2d because the alternative approach, though conceptually clearer, resulted in a significant amount of missing or invalid data due to fabrication constraints. The approach in the original manuscript, although it appears that the diameter of the lattice edges is cluttered, this is what we chose based on each length of the lattice edges so that each column (Supplementary Fig. S3) has a close relative density. Below is a detailed explanation addressing the reviewer's suggestion of fixing one parameter while varying the other:

In the new experimental set, each **row** corresponds to a fixed rod **length**, and each **column** corresponds to a fixed rod **diameter**, allowing us to analyze the individual impact of each factor. The results reveal the following:

Rod diameter and structural integrity: Structures with small diameters (e.g., D2 or D3) often failed to form complete 3D lattices, especially at longer lengths (L6, L8), due to insufficient mechanical stability during two-photon polymerization. In contrast, thicker rods (e.g., D6, D8) yielded more robust architectures with less deformation, indicating that increased diameter enhances mechanical strength.

Rod length and deformation behavior: Shorter rods (L2, L4) demonstrated relatively uniform shrinkage and swelling, while longer rods (L6, L8) were prone to asymmetric deformation or collapse, particularly when combined with thinner diameters. This highlights the trade-off between flexibility and structural integrity.

Combined effects: A balanced ratio between length and diameter is essential. Structures such as L2D6 and L4D6 exhibited the most stable and predictable actuation behavior, whereas combinations like L8D2 or L8D3 were either unstable or unmanufacturable.

Although this fixed-parameter design allows for clearer interpretation of individual influences, the number of successful fabrications is significantly reduced—particularly in cases with low diameter-to-length ratios. For example:

Fixing $D = 2$ allows only L2D2 to form a stable structure; L4D2, L6D2, and L8D2 collapse during fabrication.

Fixing $D = 6$ results in structures like L2D6 becoming nearly solid, with little meaningful deformation.

Given these constraints, we maintained the original comprehensive dataset in Fig. 2d,
which provides broader coverage of parameter space and supports more generalizable
conclusions. The supplementary figure (Fig. R1.1) complements this by illustrating the
trade-offs and design limitations more clearly.

We appreciate the reviewer's suggestion, which has helped us clarify the structural-
performance relationship and improve the presentation of our experimental
methodology.

**Fig. R1.1** Optical microscopy images of lattice structure in the swollen and shrunken
states. **a** Optical microscopy images of lattice structures of multiple parameters in the
dehydrated and shrunken state. **b** Optical microscopy images of lattice structures of
multiple parameters in the swollen state. Scale bar: 100 μm .

**The Reviewer #1's Comment 7**

Please label all the structures in Supplementary Movie 1.

**Response:**

We sincerely appreciate the reviewer's suggestion. We have updated Supplementary
Movie 1 by labeling all relevant structures to improve clarity.

**Revisions:**

We have revised Supplementary Movie 1 and the new video screenshots are below:

Fig R1.2 A frame of Supplementary Movie 1 to label all the structures.

**The Reviewer #1's Comment 8**

Do not speed up Supplementary Movie 5. The fast pace makes it difficult to observe
how the LSMR navigates through the narrow slit. A slowed-down version would be
more informative.

**Response:**

We sincerely appreciate the reviewer's suggestion. We have modified the video speed
of Supplementary Movie 5. The video of squeeze through slit of 40 μ m has been
adjusted to 0.5x speed and the video of squeeze through slit of 45 μ m has been adjusted
to 0.25x speed. We apologize for the fact that our video capture system outputs video
at a frame rate of only 15 frames per second, we have slowed down the playback as
much as possible to provide more information.

**Revisions:**

Detailed revisions are in Supplementary Movie 5.

**The Reviewer #1's Comment 9**

The authors need to provide more details on how the stiffness measurements of the
lattice blocks shown in Figure 2f and Supplementary were conducted. What was the
experimental setup used to obtain these results?

**Response:**

We appreciate the reviewer's request for additional details on the stiffness
measurements. To address this, we have revised the Supporting Information to include
Supplementary Fig. S4, which illustrates the experimental setup used for mechanical
testing of the swollen lattice hydrogel structures. The specific equipment model and
probe specifications are provided in the **Materials and Methods** section under "SEM
and mechanical characterization."

**Revisions:**

On page 6, line 151-154 of the Revised Manuscript.

We modified the manuscript to add details of the experimental setup for the stiffness
measurements as "We measured the stiffness of individual blocks using a micro
mechanical testing system. The stiffness of blocks with rod lengths of 6 μm decreased
as relative density decreased (Fig. 2f). The experimental setup and measurement
process are detailed in Supplementary Fig. S4, while additional stiffness data for
individual blocks are provided in Supplementary Fig. S5."

On page 7, line 105-118 of the supporting information.

We inserted Fig. S4 in the Supporting Information with details Modifications:

**Fig. S4. Schematic diagram of the PNIPAM-SWNT hydrogel mechanical property**
**test setup and process. (a)** Mechanical test setup. On the left side is the micro-imaging
system for real-time observation of the testing process. On the right side and below is

the illumination system. The displacement stage at the back drives the fixed arm as well
as the probe for stiffness testing. The system captures the displacement of the probe and
the force exerted by the probe in real time. **(b)** Water immersion test device. An O-ring
seal was pasted on the coverslip surface with UV-curable resin. Deionized water was
dripped inside the seal to keep the water surface level. The probe penetrated deep inside
the deionized water to measure the PNIPAM-SWNT hydrogel block in the dissolved
state. **(c)** Photograph of microscopic imaging of the measurement process. **(d)**
Schematic of the measurement process. The probe was firstly penetrated deep into the
water by visualization. Then the stiffness value brought by the substrate is measured in
the blank, and then the measured value is input into the system for correction. Finally,
each hydrogel block is measured in turn to obtain the corresponding displacement-force
data.

The Reviewer #1's Comment 10

Line 163: Please elaborate more on the rationale for selecting L4D4 as the unit structure for the LSMR.

Response:

The selection of L4D4 as the unit structure for the Lattice Soft Micro-Robot (LSMR) was based on a comprehensive evaluation of structural stability, deformation performance, and compatibility with device-scale requirements. Below, we elaborate further on the rationale behind this choice.

The truncated octahedron lattice structure utilized in LSMR is designed to achieve a balance between lightweight construction and mechanical robustness. As described in the manuscript, the lattice structure consists of hexagonal and tetragonal cells constructed using rods of uniform length (L) and diameter (D), where different combinations of L and D (e.g., $LxDy$) define the specific characteristics of each unit cell. By varying these parameters, we created lattice structures with different relative densities and porosities, which directly influenced their deformation behaviors under external stimuli such as light-induced heating or solvent evaporation.

Our experimental and simulation results demonstrated that the shrinkage ratio of the lattice structure increases as its relative density decreases. For instance, the solid structure exhibited an average linear shrinkage ratio of 15.08%, while the lattice structure achieved an average maximum shrinkage ratio of approximately 39.90% (e.g., L2D2.5, L4D4, L6D5). This reflects a significant enhancement in deformation capacity compared to the solid structure. However, there are practical limits to this enhancement. When the rod diameter becomes smaller than the rod length, the mechanical strength of the rods decreases, leading to instability and potential collapse of the three-dimensional lattice structure. This explains why the L2 series contains fewer data points compared to other series, as observed in the experiments.

Among the candidate lattice structures evaluated in our study—L2D2.5, L4D4, and L6D5—all three demonstrated impressive deformation capabilities under external stimuli such as light-induced heating or solvent evaporation. However, while L2D2.5 exhibited the highest shrinkage ratio among the three candidates, it was ultimately not selected as the unit structure due to its unsuitability for larger-scale applications.

In the L2D2.5 design, the rod diameter is smaller than the rod length, which leads to a critical limitation: the formation of near-closed cavities within the truncated octahedron lattice. While this characteristic does not significantly affect performance at smaller scales (e.g., test blocks on the order of tens of micrometers), it presents challenges when scaling up to structures of 100 μm or greater. In these cases, the reduced porosity caused by the closed cavities diminishes both the photothermal responsiveness and the overall deformation capacity of the lattice. Specifically, the internal lack of open voids in the

607 microstructure results in slower diffusion of water molecules across the hydrogel during
contraction. This limitation delays the expulsion of solvent necessary for achieving
rapid shrinking and hinders the LSMR's ability to maintain swift and robust responses
under stimulus. Thus, despite its favorable deformation properties at small scales, the
L2D2.5 configuration is unsuitable for constructing an effective soft micro-robot.

In contrast, both L4D4 and L6D5 offer greater porosity due to their larger rod diameters
relative to rod lengths. This ensures that the internal cavities remain sufficiently open,
allowing for efficient solvent exchange and maintaining high photothermal
responsiveness even at larger scales. Both configurations are therefore promising
candidates for the LSMR.

To further refine our selection between L4D4 and L6D5, we conducted a detailed
analysis of their mechanical properties, focusing on stiffness as a critical parameter for
structural stability and operational reliability. As evident from the quantitative data
presented in Supplementary Fig. S4, the L4D4 structure exhibits significantly higher
stiffness ($\sim 250 \mu\text{N}/\mu\text{m}$) compared to L6D5 ($\sim 75 \mu\text{N}/\mu\text{m}$). This marked difference in
stiffness has important implications for both structural integrity and deformation
performance.

Furthermore, the higher stiffness of L4D4 ($\sim 250 \mu\text{N}/\mu\text{m}$) ensures better structural
stability during dynamic operations. In soft microrobotic systems, maintaining
structural integrity under varying conditions is critical for consistent performance. The
balanced design of L4D4 achieves this dual objective—providing robustness against
mechanical stress while retaining sufficient flexibility for large-scale deformations.

In summary, while both L4D4 and L6D5 demonstrate notable mechanical and
deformation properties, the superior stiffness of L4D4 ($\sim 250 \mu\text{N}/\mu\text{m}$) makes it the
optimal choice for the LSMR. This decision ensures a harmonious balance between
structural stability and deformation capability, enabling the micro-robot to achieve
reliable performance in practical applications.

We hope this response addresses the reviewer's question effectively.

**Revisions:**

On page 7, line 169-177 of the Revised Manuscript.

We have modified the manuscript notes concerning the choice of lattice parameters as
“Based on the evaluation of structural stability, deformation performance, and
scalability, we selected L4D4 as the unit structure for the Lattice Soft Micro-Robot
(LSMR). Among the candidate structures (L2D2.5, L4D4, and L6D5), all demonstrated
significant shrinkage ratios under external stimuli; however, L2D2.5 exhibited
limitations in scalability due to its near-closed cavities and reduced porosity, which
hindered solvent exchange and photothermal responsiveness at larger scales. While
both L4D4 and L6D5 provided sufficient porosity for efficient deformation, L4D4

showed significantly higher stiffness ($\sim 250 \mu\text{N}/\mu\text{m}$ compared to $\sim 75 \mu\text{N}/\mu\text{m}$ for L6D5),
ensuring better structural integrity during dynamic operations. This balanced design of
L4D4 achieves optimal performance in terms of deformation capability and mechanical
robustness, making it the most suitable choice for LSMR applications.”

The Reviewer #1's Comment 11

Line 207: What are the dimensions and weight of the solid structure used in comparison with the lattice structure? Are both structures equivalent in weight and 2D dimensions?

Response:

We appreciate the reviewer's insightful question regarding the dimensions and weight comparison between the solid and lattice structures.

Dimensions Comparison: The solid structure was designed to have the same three-dimensional dimensions as the lattice structure to ensure a direct comparison.

Weight Difference: Since both structures were fabricated using the same femtosecond laser direct writing parameters, we assume the hydrogel material density remains consistent. However, due to the lower relative density of the lattice structure, its overall weight is significantly reduced compared to the solid structure.

Revisions:

On page 9, lines 222-226 of the Revised Manuscript.

We have revised the manuscript to clarify the dimensional and weight comparison between the solid and lattice structures as “To compare the movement speeds of solid and lattice structures under the same scanning speed (120 $\mu\text{m/s}$), both structures were designed with identical three-dimensional dimensions to ensure a fair comparison. Since they were fabricated under the same femtosecond laser direct writing parameters, their movement behaviors could be directly evaluated; the lattice design inherently reduces weight, which is one factor in the improved agility and efficiency observed.”

The Reviewer #1's Comment 12

Line 63: Both manual and closed-loop control systems show significant trajectory-following errors. Can the authors explain why these errors occur and suggest ways to minimize them?

Response:

We appreciate the reviewer's observation regarding trajectory-following errors in both manual and closed-loop control systems. These errors arise due to different factors in each control mode:

Manual control errors: In manual mode, the laser scanning path is predefined, and we manually adjust the piezo motorized stage to align the LSMR with the scanning trajectory. The errors in this mode primarily stem from:

1. Human judgment limitations: Since positioning adjustments rely entirely on human observation, there is an inevitable margin of error.
2. Lack of prominent reference markers: There are no clearly identified points in manual control as there are in closed-loop drives, so errors in the motion path can occur. For example, in Fig. 5b, the horizontal line in the middle of the "H" path is deviated.
3. Delayed feedback control: The absence of a real-time correction mechanism results in gradual deviation from the intended path.

Closed-loop control errors: In automated control mode, trajectory-following errors primarily arise due to coordinate transformation inaccuracies, environmental instabilities, and hardware limitations:

1. Coordinate transformation errors: The LSMR's position is first detected using image-based tracking, where coordinates are measured in pixel units. The galvanometer-driven laser scanning system operates in micrometer-scale coordinates, requiring a coordinate transformation to map the detected position onto the laser scanning path. This mapping process is not perfectly calibrated, leading to positional deviations. The issue becomes more pronounced near the field-of-view edges, where optical distortions and calibration imperfections amplify the error. Since the LSMR's movement is highly sensitive to the precise position of the laser spot, even small deviations in the scanning trajectory can significantly affect its movement direction and overall tracking accuracy.
2. Environmental instabilities: Surface imperfections on the substrate can introduce unexpected resistance, subtly affecting movement. Unsealed liquid environments and minor airflow disturbances may cause slight fluid movement, leading to unpredictable external forces on the microrobot.
3. Hardware limitations in image processing: Due to restricted access to the microscope's CCD interface, we currently rely on screen-captured video for image processing. This indirect acquisition method reduces detection accuracy, introducing

additional tracking errors. A direct hardware-integrated imaging system would improve
real-time tracking and motion control precision.

These factors collectively contribute to trajectory-following errors, and improvements
in calibration, environmental control, and direct CCD integration could further enhance
system performance.

**Potential Solutions:** To minimize these errors, the following improvements can be
considered:

1. Enhancing environmental stability: External disturbances such as vibrations and
temperature fluctuations can affect both imaging and actuation accuracy. Stabilizing
the experimental setup could improve precision.

2. Improving hardware synchronization: More precise calibration between the imaging
system and galvanometer scanner can reduce coordinate transformation errors.

3. Optimizing optical alignment: A more refined optical setup can mitigate distortions,
ensuring that the laser spot follows the intended path more accurately.

4. Direct hardware-based image processing: Instead of relying on software-based image
analysis, integrating real-time image processing into the hardware can enhance
response speed and positioning accuracy.

The Reviewer #1's Comment 13

What is the maximum hopping height achieved by the LSMR? Could the hopping motion be utilized to overcome obstacles of a certain height?

Response:

We appreciate the reviewer's interest in the hopping motion of the LSMR. Below, we clarify the nature of this motion and its potential for overcoming obstacles:

1. Clarification of hopping motion: The hopping motion observed in our experiments refers to a stepping-like movement rather than a purely vertical jump. Unlike peristaltic motion, which produces continuous forward movement, hopping occurs as a stepwise progression under high laser power.

2. Potential for overcoming obstacles: While hopping may contribute to obstacle traversal, we do not consider it the most effective approach. Supplementary Fig. S7 and Supplementary Movie S5 demonstrate a more feasible method for crossing barriers, where the LSMR deforms and actively squeezes through constrictions. Precise control over hopping height and landing position remains challenging in our current system, which is why we have not conducted extensive studies in this direction.

3. Experimental insights: We have conducted preliminary tests suggesting that hopping may facilitate obstacle crossing in some scenarios. However, due to difficulty in precisely controlling floating height and post-hopping positioning, we have prioritized alternative strategies for overcoming barriers.

The Reviewer #1's Comment 14

The discussion on future work is brief. The authors should provide more detailed elaboration on potential real-world applications of the LSMR and how they plan to address the current limitations to make the robot practical for such use cases.

Response:

We sincerely thank the reviewer for the valuable comment regarding the discussion of potential real-world applications. We appreciate the opportunity to improve the clarity and completeness of our *Discussion* section. In the revised manuscript, we have reorganized the structure of this section to provide a more coherent narrative: the first paragraph summarizes the key findings of the study, the second paragraph discusses the current limitations of the LSMR system, and the third paragraph specifically elaborates on possible application directions in light of the unique advantages of the proposed design.

In the third paragraph of the revised *Discussion*, we now provide a more detailed account of the potential applications of the Lattice Soft Microrobot (LSMR). These envisioned directions are motivated by its structural and functional features, which offer clear advantages over traditional microrobots and allow the LSMR to address several known limitations in the field.

Specifically, the low actuation energy requirement of the microlattice structure enables each LSMR unit to be activated with minimal light intensity. This feature becomes especially advantageous when multiple microrobots are operated simultaneously using beam-splitting techniques, such as spatial light modulators (SLM) or digital micromirror devices (DMD), which inherently divide the total laser power among multiple foci. Under such conditions, conventional solid microrobots often suffer from insufficient energy input at each focal point, limiting their responsiveness and the total number of controllable units. In contrast, the energy-efficient nature of the LSMR design allows a larger number of microrobots to be independently actuated under the same total laser power, thereby increasing the scalability and precision of multi-agent control systems.

Furthermore, the microlattice configuration offers a significantly larger specific surface area compared to bulk microrobots, providing more surface binding sites for loading functional agents such as drug molecules or biochemical probes. This structural advantage supports applications in targeted therapy and diagnostic sensing, where increased loading capacity can enhance therapeutic efficacy or detection sensitivity. As an illustrative example, the LSMR could be directed to a specific site within a biological environment and then activated by localized heating to trigger the release of its payload. This strategy enables spatiotemporally controlled delivery without relying on

continuous optical trapping, which can be technically demanding and potentially
damaging to surrounding tissues.

While experimental validation of these applications lies beyond the scope of the current
study, we note that relevant foundational research has been cited in our response to
Comment 5 (References 1–4), which demonstrates the feasibility of key components
such as light-triggered actuation, controlled release. Building upon this foundation, our
study aims to contribute a novel microlattice-based structural design that improves
actuation efficiency and functional integration at the microscale. We believe that the
strategies employed here—namely, enhancing material responsiveness through
microarchitectural design and achieving sophisticated functionalities using structurally
tunable soft microrobots—may offer useful insights for future developments in the field.
We hope that this revised and more detailed discussion addresses the reviewer’s
concern and highlights the forward-looking potential of our work.

**Revisions:**

On page 18, lines 415-423 of the Revised Manuscript.

We extend the discussion of potential applications of LSMR in manuscript with detailed
revisions with more subdivided areas and more detailed examples of the possible
benefits that lattice structure would bring to the design of soft microrobots.

**The structural and functional features of LSMR suggest potential in several domains.**
**First, the low actuation energy requirement of the microlattice design allows a higher**
**number of microrobots to be independently controlled under a single light source,**
**especially when using beam-splitting techniques like spatial light modulators (SLM) or**
**digital micromirror devices (DMD). This makes LSMR a promising candidate for high-**
**precision multi-agent control. Second, due to its large specific surface area, the LSMR**
**can carry a greater payload of functional agents compared to solid microrobots. This**
**feature could be advantageous in applications such as localized drug delivery or**
**biochemical sensing. For example, drugs could be released at target sites via localized**
**heating, offering spatiotemporal control without requiring constant optical trapping.**

The Reviewer 2

The authors introduce a light-actuated soft microrobot that leverages a lattice structure.

I think this work is novel, complete, and useful to the community, and will be ready for

publication after addressing a few comments:

Response:

We sincerely appreciate your positive feedback and recognition of our manuscript. We

have carefully considered your comments and have made the necessary revisions

accordingly. Below, we provide detailed responses to each of your suggestions. Thank

you again for your time and constructive feedback. We look forward to your further

review.

The Reviewer #2's Comment 1

Firstly, I think the introduction is very general, and could benefit from a more in-depth discussion of the work that immediately preceded this. Have lattices been used before to show locomotion? Has other work focused on the compression of lattices? Have other works used this principle effectively in other robots at any scale?

Response:

We sincerely appreciate the reviewer's insightful feedback regarding the depth of our Introduction. We acknowledge that the discussion on lattice-based microrobots in our manuscript was not sufficiently in-depth. Below, we provide detailed responses to each of the reviewer's comments:

1. In the field of microrobotic design, several studies have employed modular strategies using stretchable or origami-inspired structures as building blocks for robot construction. For instance, Lee et al. developed a soft robot based on a three-dimensional tensile metamaterial, which demonstrated walking locomotion.¹ Cui et al. assembled 3D robotic architectures using artificially engineered piezoelectric metamaterial lattices, enabling multi-degree-of-freedom actuation, strain amplification, and feedback control.²

In the domain of 4D-printed hydrogel lattices, Hua et al. designed a PVA/(PVA-MA)-g-PNIPAM hydrogel, utilizing 4D printing to fabricate centimeter-scale lattice structures.³ Their work demonstrated that the temperature responsiveness of the lattice-structured hydrogel was four times higher than that of bulk structures. However, this study only exhibited the bending of a bilayer beam structure and did not demonstrate untethered motion.

To the best of our knowledge, no previous study has reported a three-dimensional lattice-based microrobot at the hundred-micrometer scale that exhibits locomotion.

2. Among the existing mobile robots constructed using lattice structures, very few studies have specifically focused on compressive lattice deformations. This is primarily because volume shrinkage leading to compressive deformation is often an inherent material property rather than a design principle.

In the broader field of 4D printing using lattice structures, researchers have primarily leveraged tensile structures or multi-stable designs to achieve overall shape transformation. The main focus has often been on the programmability of unit cell structures. For instance: Zhang et al. proposed a method for multimodal programmable deformation based on a single initial structure.⁴ By adjusting the fabrication parameters of microbeams, they controlled the topological morphology of the lattice, enabling applications such as encrypted displays. Huang et al. modulated the direct writing parameters of lattice edges to achieve different deformation modes.⁵ By assembling

modular units, they demonstrated three-dimensional-to-three-dimensional structural
transformations.

However, prior work has largely overlooked the use of micro-lattice architectures in the
construction of soft microrobots, particularly with regard to enhancing both mechanical
compliance and actuation efficiency. Moreover, existing designs struggle to balance
structural programmability with high deformability in soft materials. While extremely
soft materials (e.g., fluids) can undergo large deformations, they inherently lack the
ability to retain predefined shapes. This poses a challenge in microrobotic applications,
where materials must be sufficiently compliant to deform under minimal stimuli, yet
structurally defined to support programmable actuation. Our approach addresses this
trade-off by enabling shape-definable, lattice-based soft materials with enhanced
deformability through microstructural design. Our work directly addresses this gap by
leveraging microstructural design of the constituent material to reduce actuation energy
requirements, thereby improving overall efficiency. This aligns with the grand
challenges identified for the next generation of multifunctional, power-efficient, and
compliant robotic systems, as highlighted by Yang et al.⁶

3. While the principle of using lattice structures to enhance the response performance
of hydrogels has been demonstrated in the work of Hua et al.³, their study primarily
focused on identifying this property rather than applying it to robotic actuation and
locomotion. They observed that lattice-structured hydrogels exhibit significantly faster
response speeds compared to bulk hydrogels but did not extend this concept to robotic
applications. In contrast, our work builds upon this phenomenon and fully exploits this
principle to design and fabricate a functional microrobot. By integrating a three-
dimensional lattice structure with light-responsive hydrogel materials, we achieve
controllable and multimodal locomotion at the microscale. This distinguishes our
approach from previous studies, as we not only validate the benefits of lattice structures
in enhancing actuation efficiency but also implement these advantages in a practical
robotic system, demonstrating controlled motion and adaptability in confined
environments.

We will add a description of lattice microrobotics and lattice-structured hydrogels to
the second paragraph of the Introduction, which will be used to provide an in-depth
discussion of how this work relates to the previous work.

**References:**

1. Lee, H. *et al.* 3D-printed programmable tensegrity for soft robotics. *Sci. Robot.* **5**,
eaay9024 (2020).

2. Cui, H. *et al.* Design and printing of proprioceptive three-dimensional architected
robotic metamaterials. *Science* **376**, 1287–1293 (2022).

- 3. Hua, M. *et al.* 4D Printable Tough and Thermoresponsive Hydrogels. *ACS Appl.*
*Mater. Interfaces* **13**, 12689–12697 (2021).
- 4. Zhang, M. *et al.* Hydrogel muscles powering reconfigurable micro-metastuctures
with wide-spectrum programmability. *Nat. Mater.* **22**, 1243–1252 (2023).
- 5. T.-Y. Huang *et al.* Four-dimensional micro-building blocks. *Sci. Adv.* **6**, eaav8219
(2020).
- 6. Yang, G.-Z. *et al.* The grand challenges of *Science Robotics*. *Sci. Robot.* **3**, eaar7650
(2018).

**Revisions:**

On page 2, lines 49-52 of the Revised Manuscript.

We add a discussion of research on lattice robotics and hydrogel lattice structures in
Introduction as “Typically, optimizing material composition is a common approach to
enhancing hydrogel performance. On the other hand, designing artificial
microstructures, such as lattices, has also been found to improve the response speed and
deformation range of hydrogels.^{33,35}”

The Reviewer #2's Comment 2

Additionally, the language in the introduction is self-important—phrases such as “...exhibits, by far, the fastest...”, with qualifiers like “unmatched,” “superior,” “remarkable,” etc., should be evident from the work, not your language.

Response:

We thank the reviewers for pointing out the shortcomings of the manuscript, and we have revised the language to maintain an objective academic writing style.

Revisions:

In the Revised Manuscript.

We have modified the language include the following:

1. We deleted the term “remarkable” in the abstract on page 1, line 23, as it is considered an exaggerated expression. The revised sentence is: “The LSMR achieves a continuous in situ rotation speed of 29.38^o/s,”
2. We replaced the word “fastest” with specific data in the abstract on page 1, line 24, as it is considered exaggerated and subjective language. The revised sentence is: “....., and exhibits a peristaltic locomotion speed of 15.15 $\mu\text{m/s}$ (0.14 body lengths per second).”
3. We deleted the expression “the fastest” on page 3, line 73: “The LSMR exhibits a peristaltic locomotion speed of 15.15 $\mu\text{m/s}$ (0.14 body lengths per second)”
4. We deleted the terms of “unmatched”, “superior”, “new” and synthesize other comments, we modified the sentence on page 3, line 77-80: The revised sentence is: “These advancements illustrate that the integration of lattice-based structural design and precise light-driven actuation contributes to improved energy conversion efficiency, facilitates multimodal locomotion, and enhances adaptability in constrained environments, which are critical challenges in the development of soft microrobots,”
5. We deleted the words “novel approach” “unique” on page 3, lines 67-69 because this is exaggerated and subjective language. We used more concise language to introduce our work in Introduction, the revised sentence is: “The lattice design reduces relative density, allowing for greater flexibility and faster deformation under light stimulation compared to solid microrobots made from the same hydrogel precursor material.”

The Reviewer #2's Comment 3

Figure 6h should have a counterpart in the supplementary materials that plots speed in some absolute unit (like mm/s or m/s) vs body length.

Response:

Thanks to your suggestion, we have added Fig. S12 as a counterpart to Fig.6 h in the Supporting Information with details revision.

Revisions:

In the Supporting Information with details revision.

We have added Fig. S12 as follows:

Fig. S12 Summary of reported robots plotted as the ratio of speed (µm/s) to body length as a function of length. The figure includes soft microrobots based on creeping, crawling, walking, and natural microbial peristalsis.

The Reviewer #2's Comment 4

Are there constraints to using light-based methods? I want a more in-depth discussion of possible applications that could use this robot and system, as is and in the future. How will you apply light inside the human body? Is this done already, and how? What are the “applications in medicine, engineering, and other fields,” and why is your solution the best, or even a plausible solution? Discussing and answering these important motivational questions would strengthen your work.

Response:

Thank you for the reviewer’s insightful questions, which have helped us refine our discussion and clarify the potential applications and limitations of our work. Below, we address each of the reviewer’s points in detail and highlight how we have modified the manuscript to incorporate these discussions.

1. While light-driven microrobots offer advantages such as remote, wireless control and precise, contactless actuation, they also face certain limitations. These include restricted penetration depth in biological tissues, sensitivity to environmental disturbances, and potential challenges in dynamic flow conditions. In response to these constraints, we have discussed in the revised manuscript that integrating doped magnetic nanoparticles with a magnetic field can enhance motion capabilities while retaining the benefits of light-driven control for precise displacement, manipulation, and controlled drug release. This hybrid approach aims to extend the applicability of our system beyond what is feasible with light alone.

2. We have expanded the discussion in the revised manuscript to highlight both current and future applications. In its current form, our LSMR system can be applied in microscale manipulation, micro assembly, and biomedical research, particularly for controlled transport in lab-on-a-chip systems and biological experiments outside the human body. In future applications, we envision its use in medical and engineering contexts, particularly where light transparency allows effective control. For instance, in optically transparent tissues such as the eye or certain superficial regions of the body, LSMRs could be used for precise manipulation and drug delivery. Additionally, in optical imaging-guided procedures, light not only enables visualization but could also provide simultaneous microrobot actuation for localized treatment.

3. While light penetration in biological tissues is limited, several existing biomedical technologies successfully employ light for internal procedures. Endoscopic and fiber-optic systems, for example, deliver light deep inside the body for imaging and laser surgery. Similarly, optical fibers could be adapted for microrobot control, guiding light to specific internal regions for actuation. Moreover, for applications that do not require deep penetration, such as superficial drug injection sites, our microrobot system could enable controlled, localized drug release with high precision. These considerations have

1007 been included in our revised manuscript to provide a more comprehensive discussion
of light-based actuation inside the body.

4. In our revised manuscript, we have expanded the discussion of potential applications
in medicine, engineering, and beyond. In medicine, LSMRs could be used in controlled
drug delivery, particularly in optically accessible areas, and in minimally invasive
procedures where high-precision, wireless control is needed. In engineering,
applications include microassembly and soft robotics, where programmable,
contactless actuation is advantageous. One key strength of our system is its capability
for precise, remote, and collective control—features that are challenging to achieve
with other actuation methods. Light-based control allows for simultaneous
manipulation of multiple microrobots, enabling cooperative behaviors and swarm-
based applications. These advantages, along with the potential for hybrid light-
magnetic control, support the feasibility of our approach in real-world scenarios.

To incorporate these discussions, we have revised the manuscript by elaborating on the
constraints of light-driven actuation, expanding the description of potential applications
in optically accessible environments, and emphasizing the unique advantages of our
system, particularly in contactless and collective microrobot control. We sincerely
appreciate the reviewer’s valuable feedback, which has helped us strengthen the
motivation and applicability of our work.

**Revisions:**

We revised the second paragraph in Discussion to add a discussion of the future
prospects for robotics applications

On page 18, lines 400-404 of the Revised Manuscript.

“Despite these advances, several limitations remain to be addressed in future work. The
use of purely optical actuation, while offering high spatial precision, limits robustness
in dynamically changing or optically heterogeneous environments.”

On page 18, lines 405-407 of the Revised Manuscript.

“To overcome these constraints, future implementations may incorporate magnetic
nanoparticles into the lattice structure, enabling large-scale locomotion through
magnetic fields while retaining precise positioning via localized light stimulation.”

We extend the discussion of potential applications of LSMR in manuscript with detailed
revisions with more subdivided areas and more detailed examples of the possible
benefits that lattice structure would bring to the design of soft microrobots.

On page 18, lines 415-423:

The structural and functional features of LSMR suggest potential in several domains.
First, the low actuation energy requirement of the microlattice design allows a higher
number of microrobots to be independently controlled under a single light source,

especially when using beam-splitting techniques like spatial light modulators (SLM) or
digital micromirror devices (DMD). This makes LSMR a promising candidate for high-
precision multi-agent control. Second, due to its large specific surface area, the LSMR
can carry a greater payload of functional agents compared to solid microrobots. This
feature could be advantageous in applications such as localized drug delivery or
biochemical sensing. For example, drugs could be released at target sites via localized
heating, offering spatiotemporal control without requiring constant optical trapping.

The Reviewer #2's Comment 5

The paper, overall, does not read as a story, as much of the text starts with “In Figure X, [...]” Instead of writing subplot by subplot, I recommend stating your result, then citing the figures in parentheticals. For example, instead of “Fig. 2e shows the response time and shrinkage ratio of both a lattice structure (L6D6) and a solid structure under light stimulation. The lattice structure achieved 30.58% (95,176 μm^3) volume shrinkage in just 1.46 seconds, whereas the solid structure took 2.50 seconds to shrink by only 10.55% (22,788 μm^3).”, say “The lattice structure achieved 30.58% (95,176 μm^3) volume shrinkage in just 1.46 seconds, whereas the solid structure took 2.50 seconds to shrink by only 10.55% (22,788 μm^3). (Fig. 2e)”. Additionally, instead of saying, “[result], as illustrated in Fig. X.”, say “[result] (Fig. X).” This will prevent distraction. There are many instances of both of these, but I think the manuscripts readability will improve drastically with these changes.

Response:

We sincerely appreciate this insightful suggestion. We recognize that excessive reliance on “In Figure X, [...]” constructions may fragment the narrative flow. To enhance readability, we have systematically revised such expressions across the manuscript. Now, results are stated first, with figure references placed in parentheses to ensure a more fluid and engaging presentation.

Revisions:

In the Revised Manuscript.

We have implemented the following modifications:

1. “The device and method used for producing the lattice structure are depicted in Fig. 1a.” on page 4, line 85-86.
2. “The simulation results are consistent with the experimental results, proving that our material model can well fit the deformation properties of lattice-structured hydrogel materials (Fig. 2b, c).” on page 5, line 119-121.
3. “As the relative density of the lattice structure decreases, the shrinkage ratio correspondingly increases (Fig. 2d).” on page 6, line 124-125.
4. “The lattice structure achieved 30.58% (95,176 μm^3) volume shrinkage in just 1.46 seconds, whereas the solid structure took 2.50 seconds to shrink by only 10.55% (22,788 μm^3) (Fig. 2e).” on page 6, line 142-144.
5. “Due to the sequential scanning of the laser, there is a difference in the maximum shrinkage time between the front and end of the LSMR, with the end shrinking earlier than the front in each scan cycle (Fig. 3b).” on page 8, line 183-185.

6. “In a single cycle, the microrobot’s step displacement is determined by the difference
between its forward displacement (from state i to state iii) and its backward
displacement (from state iii to state iv) (Fig. 3c).” on page 8, line 186-187.

7. “The solid structure (laser power 300 mW) and the lattice structure (laser power 50
1094 mW) traveled different distances over 14 seconds (Fig. 3d, e). (Supplementary Movie
3).” on page 9, line 226-228.

8. “A linear laser scanning path aligned with the LSMR’s body orientation (power: 60
1097 mW, scanning speed: 120 $\mu\text{m/s}$) was applied, enabling controlled movement (Fig. 4a)
1098 (Supplementary Movie 4). The superimposed image of the LSMR’s motion highlights
its trajectory (Fig. 4b), while position and timing data in the X and Y directions reveal
an average speed of 5.25 $\mu\text{m/s}$ (Fig. 4c).” on page 11, line 247-251.

9. “The trajectory induced controlled rotational motion through localized deformations
in a circular pattern (Fig. 4d). The laser rotates clockwise around the LSMR’s center of
mass, generating a rightward peristalsis at the front and a leftward peristalsis effect at
the end of the LSMR (Fig. 4e).” on page 11, line 258-261.

10. “Angular variations and center position offsets throughout the $\pm 180^\circ$ scanning
process were recorded, with an average rotational speed of 29.38 $^\circ/\text{s}$ over 360 $^\circ$ and a
maximum speed of 51.02 $^\circ/\text{s}$ (Fig. 4f).” on page 11, line 266-267.

11. “Precise trajectory control is enhanced through closed-loop feedback mechanisms
and computer vision recognition systems. the difference between open-loop and closed-
loop control for peristalsis shown in Fig. 5a.” on page 13, line 316-318.

12. “The high-power laser, directed at the tail end of the LSMR’s body, creates a
temperature gradient in the solution that drives forward movement during linear
hopping (Fig. 6a). A sequence of images from the linear hopping experiments
demonstrates this behavior (Fig. 6b, Supplementary Fig. S11, Supplementary Movie 8).
Variations in the laser’s action area led to deviations from the intended movement path,
with displacement observed in both the X and Y directions. The average movement
speed of the LSMR was 9.36 $\mu\text{m/s}$, with a maximum speed of 20.63 $\mu\text{m/s}$ (Fig. 6c).”
on page 16, line 354-359.

13. “Additionally, the forward direction can be altered by adjusting the laser’s action
area. Fig. 6d illustrates the laser’s action area during continuous hopping toward the
right front.” on page 26, line 360-365.

14. “To demonstrate continuous hopping, we navigated the LSMR through a maze
photolithographically patterned onto the substrate using UV-cured AZ-5214 photoresist,
featuring four designated turn areas along the navigation path (Fig. 6g).” on page 16,
line 374-376.

15. “Finally, the body length and movement speed of various soft-bodied robots with
peristalsis locomotion are compared (Fig. 6h and Supplementary Fig. S12).” on page
16, line 379-380.

The Reviewer #2's Comment 6

Small things: the two parts of Fig 3a could likely be combined (indicating the laser location on the top-down view).

Response:

Thank you for your valuable suggestion. We have revised Fig. 3a where the laser location is now indicated directly on the top-down view. This modification enhances clarity and conciseness, making it easier for readers to interpret the experimental setup.

Revisions:

In of the Revised Manuscript.

We modified Fig. 3a in manuscript with details revision.

Fig. R2.1 Laser position shown in top view

The Reviewer #2's Comment 7

Supplementary movies are referenced incorrectly—in supplementary materials document it says there are 10 videos, and in the text, it references video 7 when it meant video 5. Just check these over and make sure they are consistent.

Response:

We sincerely apologize for this oversight in the manuscript. We have carefully reviewed both the manuscript and the supplementary materials to ensure that all video references are correct and consistent. In the manuscript with detailed revisions, the descriptions in the text now accurately correspond to the correct supplementary videos.

Revisions:

In of the Revised Manuscript.

We have modified the following to maintain correspondence between the supplementary material and the main text:

1. “For rotational control in the X-Y plane, we employed a circular scanning trajectory tailored to the microrobot’s dimensions (Supplementary Movie 4).” On page 11, line 258.
2. “To demonstrate the LSMR’s soft and adaptable capabilities, we designed it to squeeze through a narrow slit of 45 μm , smaller than its 60 μm body width (Fig. 4g) (Supplementary Movie 5).” On page 122, line 292.
3. “To demonstrate manual control, we designed an “H” shaped path (Supplementary Movie 6).” On page 13, line 322.
4. “We further validated the effectiveness of the closed-loop drive mechanism by designing two trajectory patterns: the “Pentagram” and the “Maze” (Fig. 5c and d) (Supplementary Movie 7).” On page 13, line 329.
5. “Fig. 6e provides a sequence of images capturing the right-turn hopping process (Supplementary Fig. S8) (Supplementary Movie 8)” On page 15, line 351.

The Reviewer 3

In this article the authors proposed to propel 100 μ m microbot with light using two different kinds of motions which are peristaltic motion and a combination of Marangoni and buoyancy induced flow.

The novelty of this work is to largely increase the efficiency by using a 3D printed metamaterial approach as well as a nanotube functionalization which increase the deflection of static part as previously reported by some of the authors in reference 31.

The article is well organized and written and the results are clearly presented. They demonstrate the capacity and original contribution of the author approaches to mobile microrobot. The approach proves to be faster (but less than a factor 2) than other light induced peristaltic microrobot report in reference 9 and demonstrate the ability to switch between two motion type.

Before considering publication, I think the following question/comments should be addressed:

Response:

We sincerely appreciate your thorough review and thoughtful comments on our work. We have carefully considered all your questions and comments and have made the necessary revisions to improve the manuscript accordingly. Below, we provide detailed responses to each of your suggestions. Once again, we truly appreciate your valuable feedback and the time you have dedicated to reviewing our work. We look forward to your further insights.

The Reviewer #3's Comment 1

I think the potential application of such microrobot should be illustrated more precisely. This is crucial to compare its advantage to other technique. For example, in which scenario would it be better to use the authors proposed system than microrobot based on optical tweezers which have the same environment constrains (i.e. environment should allow light to be focus on the microrobot)?

Response:

We sincerely thank the reviewer for the insightful comments. In response, we have revised the *Discussion* section to more clearly elaborate on the potential real-world applications of our lattice soft microrobot (LSMR) and to distinguish its actuation mechanism and functional capabilities from those of conventional optical micromanipulation techniques such as optical tweezers.

1. Discussion of potential applications: We have expanded the discussion to highlight specific scenarios in which the LSMR may offer advantages. These include applications in biomedical and engineering domains where programmable, shape-morphing capabilities are desired. For instance, in biomedical contexts, the LSMR may enable targeted drug delivery in optically accessible tissues and could be integrated with optical imaging systems for simultaneous visualization and manipulation. In engineering, its ability to achieve dynamic reconfiguration and three-dimensional deformation may benefit the development of adaptive materials and reconfigurable soft robotic systems. Such applications demonstrate the potential of the LSMR for untethered, high-precision operation in transparent or confined environments, such as microfluidic lab-on-a-chip systems.

2. Comparison with optical tweezers: To further clarify the novelty of our system, we have added a direct comparison with optical tweezers. Although both systems utilize focused light and operate in optically transparent media, they differ fundamentally in actuation mechanisms. Optical tweezers rely on gradient forces to trap and manipulate small, rigid particles, whereas the LSMR is driven by structured light-induced deformation of soft lattice materials. This enables multimodal locomotion, complex shape transformations, and coordinated group control, which are difficult to realize with optical tweezers. We have added the following sentence in the revised manuscript to emphasize this distinction: *“Compared to optical tweezers, which are primarily suited for manipulating nanoscale objects and are highly sensitive to local heating and optical path quality, our system offers enhanced robustness for mesoscale object manipulation with reduced sensitivity to environmental optical heterogeneity.”*

We appreciate the reviewer’s comment, which has helped us to clearly articulate the unique advantages of our approach and its relevance to future real-world applications.

The revised manuscript now includes a more precise discussion reflecting these
important points.

**Revisions:**

On page 18, lines 402-405 of the Revised Manuscript.

We revised the second paragraph of Discussion to add a discussion of the future
prospects for robotics applications as “Compared to optical tweezers, which are
primarily suited for manipulating nanoscale objects and are highly sensitive to local
heating and optical path quality, our system offers enhanced robustness for mesoscale
object manipulation with reduced sensitivity to environmental optical heterogeneity.”

The Reviewer #3's Comment 2

I don't understand the point of the speed model presented in equation 2. Indeed the best parameters for the robot actuation seems to be set afterwards directly using an empirical approach. The goal of this model should be stated clearly. It should be compared to experiment data to be validated.

Response:

We sincerely appreciate the reviewer's thoughtful comments and recognize that our explanation of the speed model in the original manuscript was not sufficiently clear.

Below, we provide a detailed clarification and have revised the manuscript accordingly.

1. Our speed model originates from the response speed measurements shown in Fig. 2e.

The net displacement per step is calculated based on the difference between the shrinkage and swelling rates of the LSMR. The actuation time is derived from the scanning speed and the LSMR body length. In our experiments, the laser repeatedly scans along a fixed straight-line trajectory, allowing us to determine the scanning frequency from the scanning length and scanning speed. Using this information, we calculated the expected relationship between the LSMR's locomotion speed and the scanning speed, as presented in Supplementary Note S1.

2. We have already compared the calculated locomotion speed with experimental results in Supplementary Fig. S6. While our model accurately predicts the maximum locomotion speed, there is a notable discrepancy in the scanning frequency at which the peak speed occurs. This deviation arises because, in real experiments, the LSMR's peristaltic motion does not always produce an identical displacement per step, whereas our model assumes uniform step sizes.

3. The observed deviation between our model and experimental results may stem from the following factors:

At lower scanning speeds and higher laser power levels, LSMR floating occurs. Due to insufficient friction with the substrate, the microrobot undergoes repeated expansion-contraction cycles without net displacement.

At higher scanning speeds, the laser exposure time per scan is too short for sufficient heat accumulation. This prevents complete deformation, leading to inconsistent peristaltic motion, where multiple scans may be required for a single step forward. In our model, each peristaltic step contributes uniform displacement, whereas in experiments, peristaltic steps vary, particularly at high scanning speeds. This explains why the scanning frequency corresponding to the maximum locomotion speed in Fig. S6d is higher in experiments than in simulations.

The Reviewer #3's Comment 3

How the numerical different coefficients of equation 2 are found (A1,A2,xo,dx,yo,A,frac,xO1,x02,K1,K2)? Is it from an experimental fit ?

Response:

We apologize for not clearly explaining the origin of these parameters in the original manuscript.

The coefficients (A1, A2, x0, dx, y0, A, frac, x01, x02, K1, K2) in Equation 2 were obtained through an experimental fit. Specifically, they were derived by fitting the photothermal shrinkage and swelling process of the LSMR, as measured in Fig. 2e. The detailed fitting results are shown in Supplementary Fig. S6a.

Revisions:

On page 8, line 195-196 of the Revised Manuscript.

To address this lack of clarity, we have made the following modifications:

In the main text, we explicitly state that these coefficients were obtained from experimental fitting as“..., and the coefficients in Equation 2 were determined by fitting the photothermal shrinkage and swelling process of the LSMR observed in Fig. 2e.”

On page 3, line 56-58 of the Revised Supporting Information.

We provide additional clarification on the fitting process and refer to the relevant figure as “To determine the parameters of Equation S8 and S9, we performed an experimental fit of the shrinkage and swelling kinetics under laser stimulation (Fig. 2e). The resulting fitted curves and extracted coefficients are presented in Supplementary Fig. S7a.”

The Reviewer #3's Comment 4

4. I think the energy efficiency as it is defined currently in the article is not a pertinent criteria. Indeed, at this scale, the kinetic energy is negligible compared to the drag friction.

To illustrate this let assumes that the robot is a 100 μ m sphere, of density 1 travelling at 10 μ m/s in water which are conditions close to the authors robot.

In this case, assuming a Stokes law for the drag (i.e. the friction due to fluid viscosity) we obtain :

$$E_{\text{kinetic}} = 10^{-22} \text{ Joule}$$

$$F_{\text{drag}} = 10^{-11} \text{ Newton .}$$

Therefore the distance to dissipate the kinetic energy with the drag is only 10 nm. This means that an estimation of the duration when the energy of the laser is converted into kinetic energy is only 10 nanometer. After this, all the laser power transferred to the robot is dissipated in drag.

To conclude, it would make more sense to consider a power ratio (Drag force on the robot/ power of the laser) or simply the speed of the robot.

Response:

We sincerely appreciate the reviewer's insightful comments and completely agree that, at the microscale, frictional forces dominate over kinetic energy. We acknowledge that drag-induced dissipation is the primary factor affecting motion, making kinetic energy an insufficient metric for evaluating energy efficiency.

In response to this concern, we conducted additional experiments to quantify the driving force of the LSMR. We used a micro force measurement system to measure the interaction forces between the LSMR and a probe. Specifically, we positioned a microprobe at one end of the LSMR and applied laser scanning from the opposite end to induce motion. Our goal was to have the LSMR collide with the probe while simultaneously recording the force exerted.

Unfortunately, the measured data exhibited significant noise, far exceeding the expected frictional force. While fluctuations were observed in the data, the scanning frequency in our experiments was significantly higher than the observed oscillation frequency, suggesting that the fluctuations were more likely due to environmental noise rather than direct force transmission. Given that the LSMR operates in a water medium and undergoes intense photothermal effects during laser scanning, we suspect that optical and thermal disturbances contributed to the noise.

To further investigate frictional interactions, we attempted to measure the static friction between the LSMR and the substrate by pushing the microrobot with a microprobe (Fig. R3.1). In our first experiment, we used the probe to directly push the LSMR and recorded the force. As a control, we performed the same probe movement without

touching the LSMR. Surprisingly, the peak forces recorded in both cases were nearly
identical (Fig. R3.2 a and b). We repeated this experiment multiple times, consistently
obtaining similar noise-dominated results. Next, we measured the influence of
environmental factors while the probe remained stationary. Factors such as water flow
and ambient vibrations introduced noise in the probe's data even in a static state. We
conducted continuous measurements for 120 s, and the results showed that the data
fluctuations caused by environmental factors were within $\pm 20 \mu\text{N}$ (Fig. R3.2 c). This is
nearly comparable to the data range observed during the propulsion tests ($\pm 20 \mu\text{N}$).

**Fig. R3.1** Microscopic image of the process of the probe driving LSMR on the substrate
surface in an aqueous environment.

**Fig. R3.2** Measured results of LSMR friction on substrate surface in aqueous
environment. **a** Displacement and force of the probe pushing the LSMR process. **b**
Repeatable process of displacement and force of the probe pushing the LSMR process.
**c** The force when the probe is stationary in aqueous environment.

These observations suggest that the frictional forces at play are extremely small—far
below the sensitivity of our measurement system. The force probe we used has a range
of $1000 \mu\text{N}$ with a resolution in the micro-Newton range, whereas theoretical
calculations (including the reviewer's estimate) suggest that the relevant frictional
forces are on the order of 10^{-11}N , well below our system's detection limit.
In our previous experiments, however, we found that it is currently not feasible to
reliably measure the frictional forces between our microrobot and the environment.

While Stokes' law is a well-known model for estimating viscous drag in low Reynolds
number regimes, it applies primarily to spherical particles moving in a bulk fluid. Our
microrobot, however, is actuated through light-induced deformation of a hydrogel body,
and its locomotion involves repeated contact with a solid silica substrate. Based on our
observations, we believe that the dominant interaction affecting motion is not fluid drag,
but rather the solid-solid friction at the interface between the NIPAM hydrogel and the
silica substrate.

At present, we have not identified a robust theoretical model to describe the frictional
interactions occurring at this hydrogel–substrate interface under light-induced cyclic
deformation. The friction force in our system is affected not only by material properties
but also by the structural design of the robot (lattice microstructure) which further
complicates modeling. Additionally, several important parameters, such as contact
force or real-time contact area, are extremely challenging to measure accurately at this
microscale.

Given the considerations above, we ultimately chose to use kinetic energy and laser
input energy—two parameters that are relatively easy to obtain from our experiments—
for comparative analysis. While we acknowledge that these parameters do not fully
reflect the complexity of microscale energy conversion, they offer a practical way to
characterize the actuation behavior under our current experimental constraints. We
hope that this approach may provide a modest reference for future studies on light-
driven microrobots, particularly in contexts where direct force or power measurements
are difficult to implement. In our view, driving efficiency is one of the important aspects
to consider when optimizing microrobot design and actuation strategies.

We fully acknowledge that a power-based metric, such as the ratio of drag force to laser
input power, would be a more appropriate indicator from a theoretical standpoint. In
future work, if we succeed in building a more stable 3D actuation environment—such
as one utilizing a combination of Marangoni flows and buoyancy-induced flows in
water—we will seriously consider adopting such approach. In such a system, the use of
Stokes' law to calculate hydrodynamic drag and compare it with laser power would be
far more applicable.

We are grateful to the reviewer for this thoughtful and constructive suggestion, which
has deepened our understanding of the limitations of current metrics and inspired
directions for improvement in future work.

The Reviewer #3's Comment 5

The rotation ability of the robot could be better presented to understand its angular precision. Figure 4 f display the rotation with time but there is no information on the angular precision that the robot can reach. Could this somehow be tuned by the laser power (or other parameters)? I think this is a crucial parameter to assess the precision the microrobot could perform in path following.

Response:

We sincerely appreciate the reviewer's insightful comment, which has prompted us to further analyze and quantify the angular precision of the LSMR's rotational motion.

In the context of our study, we define angular precision as the average angular displacement per laser scanning cycle during rotation. While we initially measured rotation speed over time (Fig. 4f), we had not explicitly presented the angular displacement per cycle. To address this, we have now included statistical measurements of the per-cycle rotation angle corresponding to the conditions used in Fig. 4f.

Motivated by the reviewer's suggestion, we conducted additional experiments to determine the minimum achievable rotational step size and investigate the effect of laser power on angular precision. For these experiments, we maintained a relatively low circular scanning speed to ensure finer resolution in measuring per-cycle angular displacement. The results are summarized as follows:

Laser power(mW)	Average angle of rotation (°)	Standard Deviation (°)
20	-	-
30	3.87	1.49
40	6.21	2.80
50	6.59	3.62

At a laser power of 20 mW, the LSMR did not exhibit any rotational motion. This is because the thermal energy generated at this power level was insufficient to induce significant deformation in the hydrogel, and thus no displacement occurred. When the laser power was increased to 30 mW, the LSMR was able to rotate, with an average angular displacement of $3.87^\circ \pm 1.49^\circ$ per complete laser scan. However, the rotation at this power level was susceptible to environmental disturbances; even minor surface impurities on the substrate could obstruct motion. This instability is attributed to the relatively weak photothermal stimulus, which resulted in limited deformation in the irradiated regions and therefore only small driving forces and displacement. Although actuation became possible at 30 mW, the motion remained unstable and sensitive to external conditions.

At 40 mW, the average rotation angle increased to $6.21^\circ \pm 2.80^\circ$, indicating that the
laser power was sufficient to induce substantial and consistent deformation of the
LSMR body, enabling more stable in-place rotation. Further increasing the power to 50
1435 mW yielded an average rotation angle of $6.59^\circ \pm 3.62^\circ$, a modest gain of only 0.38° ,
suggesting that actuation performance had reached a saturation point at 40 mW. At 50
1437 mW, the expanded thermal influence zone caused nearly half of the LSMR body to
1438 undergo contraction during circular scanning, resulting in a larger standard deviation of
1439 rotation angle and occasional translational displacement during rotation. This illustrates
that while higher laser power can enhance deformation, it may also compromise
rotational precision due to increased variability and off-center motion.

These results demonstrate that while angular displacement can be tuned by adjusting
laser power, precise control requires a balance between actuation intensity and
deformation stability.

**Revisions:**

On page 12, line 278-290 of the Revised Manuscript.

We added the results of the study on angular precision and laser power as “To quantify
the angular precision of the LSMR, we conducted experiments using laser powers of
20mW, 30 mW, 40 mW, and 50 mW at a scanning speed of $120 \mu\text{m/s}$ (Supplementary
Fig. S8). The angular displacement per complete circular laser scan increased with laser
power, averaging $3.87^\circ \pm 1.49^\circ$, $6.21^\circ \pm 2.80^\circ$, and $6.59^\circ \pm 3.62^\circ$, respectively. No
rotation was observed at 20 mW, as the thermal input was insufficient to induce
effective deformation of the hydrogel structure, resulting in a lack of actuation force
and thus no measurable rotation angular. At 30 mW, the LSMR began to rotate, though
the motion remained unstable and was easily impeded by surface irregularities due to
the limited deformation and actuation force. At 40 mW, a more robust and consistent
rotation was achieved, suggesting that the deformation had reached an effective
threshold. Further increasing the power to 50 mW resulted in only a slight gain in
rotation angle but introduced larger variability, likely due to the expanded thermal
influence zone and less localized actuation. These results demonstrate that rotational
behavior is tunable by laser power, with a trade-off between actuation intensity and
motion stability that must be optimized for precise path-following applications.”

On page 9, line 135 of the Revised Supporting Information.

We have added a data figure on the experimental results of angular precision and laser
power.

**Fig. S8** The mean angular precision of LSMR under different laser power levels.

Angular precision is defined as the angle of rotation of the LSMR per laser scan rotation.

The Reviewer #3's Comment 6

Could you provide details on the method to adjust direction in the closed loop control scheme as display in Fig S8?

Response:

Thank you for your question. Below, we provide a detailed explanation of the direction adjustment process in the closed-loop control scheme. First, we use Python-OpenCV to extract the contour of the LSMR and fit it with a minimum bounding rectangle, as shown by the green rectangular box in Fig. R3.1. We then obtain the center point of the rectangle and the angle between its longer side and the vertical axis. We define the center of the fitted rectangle as the origin, with the upward vertical direction as 0° , as shown by the orange line in Fig. R3.1. Since the LSMR is a symmetric structure, we do not distinguish between its head and tail, allowing us to unify its orientation within the range of 0° to 180° . The LSMR angle is determined by rotating clockwise from the 0° reference line until it aligns with the longer side of the rectangle.

Next, we determine the target point angle by constructing a line from the rectangle's center to the target point. The target angle is defined by rotating clockwise from the 0° reference line until it aligns with this constructed line. Similar to the LSMR angle, the target angle is also unified within the 0° to 180° range. The angle difference between the target point and the LSMR is then computed. If the angle difference is less than 90° , the laser is controlled to scan clockwise, as illustrated in Fig. R3.1b and c. If the angle difference is greater than 90° , the laser is controlled to scan counterclockwise, as shown in Fig. R3.1a and d. This method ensures precise directional adjustments in the closed-loop control scheme.

**Fig. R3.3** detailed method to adjust direction in the closed loop control scheme. The
 green rectangles represent the outlines of the LSMRs recognized by python-based
 OpenCV. The blue points represent the artificially set target points. The orange line
 represents the artificially set 0-degree line used for reference. The yellow line represents
 the direction of the long side of the rectangle. The purple line represents the line
 between the target point and the center of the rectangle. Angle_rec represents the angle
 between the long side of the rectangle and the 0 degree line at this point in time.
 Angle_target represents the angle between the line between the center of the rectangle
 and the target point and the 0 degree line. Dist represents the distance between the center
 of the rectangle and the target point, in pixels. Diff represents the angle between the
 direction of the long side of the rectangle and the target point. **a b c** and **d** correspond
 to different cases where the target point is in the center of the rectangle in each of the
 four directions.

The Reviewer #3's Comment 7

In general the closed loop is very crude and could inspire from other closed loop in the microrobotics literature. More detail on how to improve it should be considered. Please find in the following an article illustrating few methods :

Dahroug, Bassem, et al. "Some examples of path following in microrobotics." 2018 International Conference on Manipulation, Automation and Robotics at Small Scales (MARSS). IEEE, 2018.

Response:

We sincerely thank the reviewer for the valuable suggestion and for recommending the paper "*Some examples of path following in microrobotics*" (Dahroug et al., MARSS 2018). This work introduces three commonly used control strategies for microrobotic motion: **Waypoint Sequence**, **Trajectory Tracking**, and **Path Following**. We have carefully studied these methods and provide a brief summary below:

1. **Waypoint Sequence**: In this strategy, the robot navigates through a discrete set of predefined waypoints, typically stopping at each point to correct its position and orientation before proceeding to the next. It is often used when feedback is only available intermittently, and when motion is performed in distinct steps. This approach is simple to implement and effective for robots with limited or discontinuous locomotion capabilities.

2. **Trajectory Tracking**: This method involves commanding the robot to follow a pre-defined time-dependent trajectory, continuously adjusting its position and orientation based on real-time feedback. It requires high temporal resolution and precise feedback control. Trajectory tracking is best suited for robots with continuous locomotion and minimal delays in sensing and actuation.

3. **Path Following**: In path following, the robot is required to follow a spatial path (typically a curve) regardless of timing. The controller ensures that the robot stays on the path geometry while allowing some flexibility in speed. This approach is particularly useful for robots capable of smooth, continuous motion, and benefits from real-time localization to maintain path adherence.

In our manuscript, we employ a closed-loop feedback control based on the Waypoint Sequence method. After analyzing the three strategies, we chose that Waypoint Sequence is best suited to our microrobot system for the following reasons:

1. **Locomotion Characteristics**: Our light-driven soft microrobot exhibits stepwise locomotion. Its turning and straight movements are decoupled and must be executed sequentially. Unlike magnetic helical swimmers or focused laser spots, our microrobot cannot perform continuous, smooth trajectory curves. Instead, it must first rotate to the target direction, then translate forward. This sequential mode of movement aligns

naturally with the waypoint sequence strategy, while trajectory tracking or path
following are difficult to realize with such discrete motions.

2. Feedback Timing Constraints: Our system requires the microrobot to pause
intermittently for image-based localization due to significant shape deformation during
actuation. During contraction, the robot's geometry changes substantially, making real-
time detection of its position and orientation unreliable. Only when the robot is at rest
can accurate feedback be obtained. This periodic feedback structure fits well with the
waypoint-based control approach, where localization and re-orientation occur at
discrete steps.

3. Control Efficiency and Precision: In path following, a dense set of reference points
is often needed to approximate the curve, which in our system would require frequent
angular adjustments. Our microrobot has an experimentally determined angular
resolution of $6.21^\circ \pm 2.80^\circ$, which limits the precision of its directional control. Too
many control points can lead to redundant corrections and inefficient navigation, a
situation we have observed in our experiments. Waypoint sequence control allows us
to limit the number of target points, optimizing travel time and improving overall
robustness.

Compared with the error introduced by the control algorithm, we believe that the
systematic error introduced by the hardware system in the existing system may be more
significant, and optimization from the following aspects may bring more obvious
results:

1. Enhancing environmental stability: External disturbances such as vibrations and
temperature fluctuations can affect both imaging and actuation accuracy. Stabilizing
the experimental setup could improve precision.

2. Improving hardware synchronization: More precise calibration between the imaging
system and galvanometer scanner can reduce coordinate transformation errors.

3. Optimizing optical alignment: A more refined optical setup can mitigate distortions,
ensuring that the laser spot follows the intended path more accurately.

4. Direct hardware-based image processing: Instead of relying on software-based image
analysis, integrating real-time image processing into the hardware can enhance
response speed and positioning accuracy.

We are grateful to the reviewer for highlighting this important aspect of closed-loop
control. While our current implementation favors waypoint sequence control, we
recognize the value of more advanced strategies such as path following. In future work,
we aim to enhance our localization techniques to enable continuous pose tracking, at
which point path-following strategies may become more feasible and beneficial for our
system.

The Reviewer #3's Comment 8

Does a femtosecond laser is required to actuate the robot (as in this presented work)?
Would it work with a continuous laser? I think this is an important information to assess
the complexity of powering such microrobot.

Response:

We sincerely appreciate the reviewer's insightful question. We agree that a continuous-wave (CW) laser would be a more practical option for photothermal actuation in many applications, as it provides a steady energy input and simplifies the system hardware.

The use of a femtosecond laser in our study was primarily due to limitations in our available experimental system. Our optical setup was originally designed for multiphoton fabrication and manipulation, which make it easy for us to apply femtosecond laser.. However, the actuation mechanism itself does not strictly require femtosecond pulses; it relies on the photothermal effect, which can also be achieved with a continuous-wave laser at an appropriate wavelength and power.

In principle, a CW laser could also effectively induce the photothermal response necessary for LSMR motion. Given the strong absorption of our material in the near-infrared (NIR) region, a CW laser with a sufficiently high power density should be able to achieve similar results, potentially even at lower peak power levels than a pulsed laser. The specific power requirements would depend on factors such as absorption efficiency, heat dissipation, and laser spot size.

Revisions:

On page 18, line 411-414 of the Revised Manuscript.

We add a discussion of femtosecond lasers and continuous lasers as “It is also worth noting that although femtosecond lasers were used in this study, this choice was driven by equipment availability. In principle, the photothermal actuation of the hydrogel can be achieved using continuous-wave lasers, which may facilitate broader adoption of the technology in practical applications.”

The Reviewer #3's Comment 9

9. The authors state: “Manual annotation of target waypoints ensures accurate navigation.”

Could you elaborate on what this sentence mean ?

Response:

We sincerely apologize for the lack of clarity in our original statement. What we intended to convey is the role of redundant waypoint annotations in improving the precision of LSMR’s motion path, as demonstrated in Fig. 5d.

In contrast to Fig. 5c, where only key waypoints at turning points are marked, LSMR relies on closed-loop feedback to navigate the path. However, due to environmental factors and system precision limitations, unexpected deviations in the trajectory may occur. For example, in Fig. 5c, the motion from waypoint 4 to 5 does not follow a perfectly straight path.

By introducing redundant target waypoints, as shown in Fig. 5d, we can guide the LSMR’s trajectory more accurately, minimizing deviations and ensuring smoother navigation.

Revisions:

On page 14, line 336-338 of the Revised Manuscript.

To clarify this point in the manuscript, we have revised the sentence as “**The use of redundant waypoint annotations allows for more precise control of the LSMR’s trajectory, reducing deviations caused by environmental and system precision factors, as demonstrated in Fig. 5d.**”

**Manuscript Revision Details**

Journal: Nature Communications

Ms. No.: NCOMMS-24-79506A

Ms. Title: Light-driven Lattice Soft Microrobot with Multimodal Locomotion

Authors: Mingduo Zhang^{1†}, Yuncheng Liu^{1†}, Chunsan Deng¹, Xuhao Fan¹, Zexu
Zhang¹, Shaoxi Shi¹, Fayu Chen¹, Huace Hu¹, Songyan Xue¹, Leimin Deng^{1,2}, Lige Liu³,
Tao Sun^{3,4}, Hui Gao^{1,2}, Wei Xiong^{1,2*}

Dear editors and reviewers,

The authors would like to thank the editor and all reviewers for their constructive
comments and suggestions for our manuscript. We are glad to get all your positive
assessments of our submission and appreciate for giving us the opportunity to submit a
revised draft. We have incorporated the suggestions made by the reviewers and revised
the manuscript accordingly.

Below, we provide a point-by-point response (in blue) to the reviewer's comments (in
black) and point out the places where we have revised. The modifications have also
been highlighted in red color in the revised manuscript.

Thanks again for your time in reviewing our manuscript. Looking forward to hearing
from you soon.

Yours sincerely,

Wei Xiong (on behalf of all authors)

weixiong@hust.edu.cn

**Revision Summary**

**Manuscript:**

- 1. Page 1, line 4, Added affiliation for author Tao Sun.
- 2. Page1, line 9, Modified affiliation information.
- 3. Page 1, line 10, Added affiliation for author Tao Sun.
- 4. On Page 2, line 29, We modified the description of energy conversion efficiency in
the abstract.
- 5. On Page 3, line 76, We modified the description of the energy conversion efficiency
in Introduction.
- 6. On Page 8, line 199, We have added an additional note on the significance of the
parameters in the fitted function.
- 7. On Page 9, line 233, We removed the description of the energy conversion efficiency
in Results.
- 8. On Page 14, line 338, We add a note on the details of closed-loop feedback.
- 9. On Page 18, line 399, We modified the description of the energy conversion
efficiency in Discussion.
- 10. On Page 18, line 410, We have added descriptions about the applicability of
waypoint sequence navigation.
- 11. On Page 16, line 358, Updated the serial numbers of the supplementary figure.
- 12. On Page 16, line 373, Updated the serial numbers of the supplementary figure.
- 13. On Page 16, line 382, Updated the serial numbers of the supplementary figure.

**Supplementary information:**

- 1. Page 1, line 4, Added affiliation for author Tao Sun.
- 2. Page1, line 9, Modified affiliation information.
- 3. Page 1, line 10, Added affiliation for author Tao Sun.
- 4. Page 1, line 14, Updated the serial numbers of the supplementary note.
- 5. Page 1, line 15, Modified the number of supplementary figure.
- 6. On Page 3, line 51, We add a description of the physical meaning of each parameter
in the fitting function for the contraction process.
- 7. On Page 3, line 61, We add a description of the physical significance of each
parameter in the fitting function for the swelling process.
- 8. On Page 3, line 84, We removed the description of the energy conversion efficiency.
- 9. On Page 12, line 138, We have added detailed information on angle recognition in
closed-loop control systems.
- 10. On Page 13, line 166, Updated the serial numbers of the figure.
- 11. On Page 13, line 169, Updated the serial numbers of the figure.

**Response to reviewer comments**

**Reviewer: 1**

The authors have carefully addressed all my comments and concerns. I appreciate the
efforts from the authors for the careful revision.

**Response:** We sincerely appreciate your positive feedback and constructive comments
during the review process. Your insightful suggestions have significantly improved the
quality of our manuscript, and we are grateful for your time and expertise in guiding
our revisions.

Thank you once again for your valuable contributions to our work.

The Reviewer 2

Thank you for thoroughly addressing my concerns!

Response: We sincerely appreciate your time and valuable feedback throughout the review process. Your constructive comments have greatly enhanced the quality of our manuscript. Thank you for your thorough and encouraging assessment.

The Reviewer 3

The authors have addressed well most of the points I previously mentioned and the overall quality of the manuscript is greatly improved. I still have one major point and three minor points before considering publications.

Response: We sincerely appreciate your recognition of the manuscript improvements and your continued engagement with our work. Your expertise has been invaluable in enhancing this study, and we are grateful for the opportunity to further refine the paper based on your latest suggestions. Below we provide point-by-point responses to your remaining concerns.

The Reviewer #3's Comment 1

It is still unclear how the 4 curves of figure S6a are used to fit all the following parameters (A_1 , A_2 , x_0 , dx , y_0 , A , frac , x_{01} , x_{02} , K_1 , K_2) in Equation 2.

The fitting methods of each parameters need to be explained. This is crucial to achieve the link between the material property on the robot speed modulation. Without it the model seems useless to others researcher.

Response: During the shrinkage process, four parameters were used in the Boltzmann fitting:

$$f_s = A_2 + \frac{A_1 - A_2}{1 + e^{t-x_0/dx}}$$

$A_1 = 0.00975$, $A_2 = 0.2027$, $x_0 = 0.33745$, and $dx = 0.1106$.

Here, A_1 represents the theoretical shrinkage ratio at the initial state (baseline level), which is very close to zero, consistent with the physical expectation that no significant shrinkage occurs at the beginning. A_2 corresponds to the final stable shrinkage plateau value, matching the experimentally observed terminal shrinkage ratio (0.2027). x_0 denotes the center time of the shrinkage transition, that is, the inflection point where the shrinkage accelerates and then slows down, indicating a very fast hydrogel response. dx is the scale factor describing the steepness of the transition; the fitted value of 0.11 s suggests a relatively rapid change from fast shrinkage to the stable state.

In our study, we selected the Boltzmann function to fit the hydrogel's volume change kinetics under light stimulation because this function provides a classical mathematical model for S-shaped (sigmoidal) dynamic processes, which corresponds closely to the physical behavior of light-driven hydrogel contraction. The Boltzmann function describes three distinct phases:

(1) Initial slow phase: When the laser stimulation is first applied, local photothermal conversion begins, gradually accumulating heat at the irradiated region. This corresponds to the initial gentle slope of the Boltzmann curve, as thermal energy accumulates but the temperature has not yet reached the phase transition threshold needed for significant volume contraction.

(2) Rapid transition phase: Once the local temperature exceeds the phase transition temperature, the hydrogel network begins to collapse, leading to rapid volume contraction. At this point, heat conduction accelerates the temperature rise in adjacent regions, triggering phase transitions over a larger area. This corresponds to the steep portion of the Boltzmann curve, where the volume change rate reaches its peak.

(3) Final stable phase: Eventually, the system reaches thermal equilibrium: the entire hydrogel block has undergone the phase transition, and the contraction stabilizes at a plateau value. This matches the flat tail of the Boltzmann curve, where the deformation saturates and further change is negligible.

We performed iterative fitting in Origin, using the Levenberg-Marquardt optimization
algorithm. This algorithm combines the gradient descent and Gauss-Newton methods,
making it well suited for nonlinear curve fitting of sigmoidal functions. It ensures a
balance between convergence speed and stability, allowing us to obtain reliable
parameter estimates that accurately reflect the hydrogel's physical response.

During the swelling process, seven parameters were used in the double Boltzmann
fitting:

$$136 \quad f_d = y_0 + A \left(\frac{frac}{1 + e^{t-x_{01}/k_1}} + \frac{1 - frac}{1 + e^{t-x_{02}/k_2}} \right)$$

$y_0 = -0.0136$, $A = 0.21649$, $frac = 0.56217$, $x_{01} = 0.74557$, $x_{02} = 2.19965$, $k_1 = -0.36545$,
and $k_2 = -0.09528$.

y_0 represents a small initial baseline offset, close to zero, which indicates a correction
to the starting point, possibly due to baseline measurement error or a slight shift in the
fitting. A represents the overall swelling amplitude (i.e., the edge length change relative
to the fully swollen state), reflecting the maximum deformation as the hydrogel
transitions from the contracted state to the fully swollen state. The final plateau value
is approximately $A + y_0 (\approx 0.21649 - 0.0136 \approx 0.2027)$, consistent with the stable
swelling condition. $frac$ indicates the contribution of the fast swelling component,
accounting for 56.2% of the total swelling amplitude, suggesting that the majority of
swelling is dominated by the fast stage. x_{01} is the center time of the fast swelling phase,
representing the characteristic time scale of this rapid transition. x_{02} is the center time
of the slow swelling phase, indicating the inflection point of the slower swelling
component. k_1 reflects the steepness of the fast stage; the fitted value of approximately
151 -0.37 s indicates that the major swelling occurs within the first ~ 1.5 seconds. The
152 negative sign is a conventional feature of the Boltzmann formulation (indicating
direction of change). k_2 corresponds to the steepness of the slow stage; although the
fitted value (about -0.095 s) implies a relatively sharp transition, the large x_{02} value
means the change actually occurs more gradually in the later phase. The double
Boltzmann form was chosen because it provides a better fit to the experimental data,
with an R^2 of 0.99925 for the double Boltzmann model, compared to 0.99245 for the
single Boltzmann model, as shown in Fig.S1 b and c.

The double Boltzmann function fitting captures the distinct stages of this process, which
can be mapped to physical phenomena as follows:

(1) Early swelling component: As cooling proceeds, the hydrogel surface cools rapidly,
enabling quick water uptake and network re-expansion near the outer regions. The first
Boltzmann component (centered at $x_{01} \approx 0.75$ s, steepness $k_1 \approx -0.37$ s) models this early

swelling behavior. The rise in the curve reflects the main fraction of swelling (about
56%, as given by frac) that occurs during this stage.

(2) Late swelling component: Subsequently, the hydrogel's internal regions, which cool
more slowly due to limited heat diffusion, continue to absorb water and adjust their
structure. This corresponds to the second Boltzmann component (centered at $x_{02} \approx 2.20$
169 s, with steepness $k_2 \approx -0.095$ s). The gradual rise in this portion of the curve reflects
additional minor swelling and final structural relaxation as the system approaches
equilibrium.

(3) Final plateau: Eventually, the hydrogel reaches a stable, fully swollen state where
no further significant volume change occurs. This corresponds to the flat plateau region
of the double Boltzmann curve, at a value of $A + y_0 \approx 0.203$, consistent with the
experimentally observed final deformation ratio.

The fitting method for the swelling data was the same as that used for the contraction
process: we performed iterative fitting in Origin, using the Levenberg-Marquardt
optimization algorithm. This algorithm combines the gradient descent and Gauss-
Newton methods, making it well suited for nonlinear curve fitting of sigmoidal
functions. It ensures a balance between convergence speed and stability, allowing us to
obtain reliable parameter estimates that accurately reflect the hydrogel's physical
response.

**Fig.S1 Fitting results of the deformation of the lattice hydrogel block with laser**
**stimulation on and off as a function of time. a** Relationship between the proportion
of hydrogel edge deformation with time starting from laser stimulation during
contraction. Based on Boltzmann function fitting results; **b** Relationship between the
proportion of hydrogel deformation with time from laser off during dissolution. Based
on the results of the double Boltzmann function fitting; **c** The variation of the proportion
of hydrogel deformation with time from the laser off during the swelling process. Based
on Boltzmann function fitting results.

**Revisions:**

On Page 8, line 199-200 of the Revised Manuscript.

We have added an additional note on the significance of the parameters in the fitted
function as “A detailed discussion of the parameter sources and their physical
implications is provided in Supplementary Note S1.”

On Page 3, lines 51-58 of the Revised Supporting Information.

We add a description of the physical meaning of each parameter in the fitting function
for the shrinkage process as “Here, A_1 represents the theoretical shrinkage ratio at the
initial state (baseline level), which is very close to zero, consistent with the physical
expectation that no significant shrinkage occurs at the beginning. A_2 corresponds to the
final stable shrinkage plateau value, matching the experimentally observed terminal
shrinkage ratio (0.2027). x_0 denotes the center time of the shrinkage transition, that is,
the inflection point where the shrinkage accelerates and then slows down, indicating a
very fast hydrogel response. dx is the scale factor describing the steepness of the
transition; the fitted value of 0.11 s suggests a relatively rapid change from fast
shrinkage to the stable state.”

On Page 3, lines 61-77 of the Revised Supporting Information.

We add a description of the physical significance of each parameter in the fitting
function for the swelling process as “Here, y_0 represents a small initial baseline offset,
close to zero, which indicates a correction to the starting point, possibly due to baseline
measurement error or a slight shift in the fitting. A represents the overall swelling
amplitude (i.e., the edge length change relative to the fully swollen state), reflecting the
maximum deformation as the hydrogel transitions from the contracted state to the fully
swollen state. The final plateau value is approximately $A + y_0$ ($0.21649 - 0.0136 \approx$
0.2027), consistent with the stable swelling condition. frac indicates the contribution of
the fast swelling component, accounting for 56.2% of the total swelling amplitude,
suggesting that the majority of swelling is dominated by the fast stage. x_{01} is the center
time of the fast swelling phase, representing the characteristic time scale of this rapid
transition. x_{02} is the center time of the slow swelling phase, indicating the inflection
point of the slower swelling component. k_1 reflects the steepness of the fast stage; the
fitted value of approximately -0.37 s indicates that the major swelling occurs within the
first ~1.5 seconds. The negative sign is a conventional feature of the Boltzmann
formulation (indicating direction of change). k_2 corresponds to the steepness of the slow
stage; although the fitted value (about -0.095 s) implies a relatively sharp transition, the
large x_{02} value means the change actually occurs more gradually in the later phase. All
fitting parameters were determined through iterative fitting in Origin using the
Levenberg-Marquardt algorithm, which combines gradient descent and Gauss-Newton
methods.”

The Reviewer #3's Comment 2

I appreciate the effort of the authors to try to measure force apply by their robot which is truly a great challenge at this scale.

Nonetheless I still think that comparing the speed (or the speed in body length per second) is more relevant than an energy efficiency based on kinetic energy. I would therefore suppress the line mentioning kinetic energy.

Response: We thank the reviewer for this valuable comment. We agree with the reviewer's suggestion and have removed the discussion on energy conversion efficiency from the manuscript. Instead, we now directly describe the difference in laser energy input and motion speed between the solid-structure and lattice-structure microrobots under identical conditions.

Revisions:

On Page 2, lines 29-30 of the Revised Manuscript.

We have removed the statement on kinetic energy-based efficiency and revised the text to focus on the locomotion performance under identical conditions as “**Compared to solid microrobot, the lattice microrobot requires only one-sixth of the laser energy to achieve three times the motion speed, under otherwise identical conditions.**”

On Page 3, lines 76-78 of the Revised Manuscript.

We have removed the statement on kinetic energy-based efficiency and revised the text to focus on the locomotion performance under identical conditions as “**Compared to solid microrobot, the lattice microrobot requires only one-sixth of the laser energy to achieve three times the motion speed, under otherwise identical conditions.**”

On Page 9, line 233 of the Revised Manuscript.

We removed the description of the energy conversion efficiency in Results-Linear peristalsis by scanning frequency modulation as “**The energy conversion efficiency of the lattice structure was 16.49 times higher than that of the solid structure, based solely on the conversion of laser energy into kinetic energy (Supplementary Note S2).**”

On Page 18, lines 399-402 of the Revised Manuscript.

We removed the statement about energy conversion efficiency in Discussion and modified it as “**Compared to solid microrobot, the lattice microrobot requires only one-sixth of the laser energy to achieve three times the motion speed and enables to squeeze through confined spaces by adapting frictional interactions with its environment.**”

On Page 3, line 81 of the Revised Supporting Information.

271 We removed the calculation procedure for the energy conversion efficiency.

The Reviewer #3's Comment 3

Thank you for the detail on the closed loop, I think these should be summarized in the article or supplementary material.

Response: We thank the reviewer for their valuable suggestion. As recommended, we have added a summary of the closed-loop details to the supplementary material to improve clarity and completeness.

Revisions:

On Page 14, lines 338-339 of the Revised Manuscript.

We supplement the information provided in the supporting material with a detailed methodology for direction determination in the closed-loop feedback process as “Detailed angle determination methods are provided in Supplementary Fig. S11.”

On Page 12, lines 138-164 of the Revised Supporting Information.

We have added detailed information on angle recognition in closed-loop control systems as “

Fig. S11 detailed method to adjust direction in the closed loop control scheme. The green rectangles represent the outlines of the LSMRs recognized by python-based OpenCV. The blue points represent the artificially set target points. The orange line represents the artificially set 0-degree line used for reference. The yellow line represents the direction of the long side of the rectangle. The purple line represents the line between the target point and the center of the rectangle. Angle_rec represents the angle between the long side of the rectangle and the 0 degree line at this point in time. Angle_target represents the angle between the line between the center of the rectangle

and the target point and the 0 degree line. Dist represents the distance between the center
of the rectangle and the target point, in pixels. Diff represents the angle between the
direction of the long side of the rectangle and the target point. a b c and d correspond
to different cases where the target point is in the center of the rectangle in each of the
four directions.

First, we use Python-OpenCV to extract the contour of the LSMR and fit it with a
minimum bounding rectangle, as shown by the green rectangular box. We then obtain
the center point of the rectangle and the angle between its longer side and the vertical
axis. We define the center of the fitted rectangle as the origin, with the upward vertical
direction as 0° , as shown by the orange line. Since the LSMR is a symmetric structure,
we do not distinguish between its head and tail, allowing us to unify its orientation
within the range of 0° to 180° . The LSMR angle is determined by rotating clockwise
from the 0° reference line until it aligns with the longer side of the rectangle.

Next, we determine the target point angle by constructing a line from the rectangle's
center to the target point. The target angle is defined by rotating clockwise from the 0°
reference line until it aligns with this constructed line. Similar to the LSMR angle, the
target angle is also unified within the 0° to 180° range. The angle difference between
the target point and the LSMR is then computed. If the angle difference is less than 90° ,
the laser is controlled to scan clockwise, as illustrated in Fig. S11 b and c. If the angle
difference is greater than 90° , the laser is controlled to scan counterclockwise, as shown
in Fig. S11 a and d. This method ensures precise directional adjustments in the closed-
loop control scheme.”

The Reviewer #3's Comment 4

Thank you for putting in perspective your closed loop with existing one in the literature. I think the explanation for choosing waypoint sequence is valid. The article would greatly benefit a sentence or two to justify the explanation of the waypoint method compared to others in this particular case.

Response: We thank the reviewer for this valuable suggestion. As recommended, we have added sentences in the main text to further justify our choice of the waypoint method and to clarify how it compares to alternative approaches in this particular case. We believe this addition improves the clarity and completeness of the manuscript.

Revisions:

On Page 18, lines 410-419 of the Revised Manuscript.

We modified the discussion of closed-loop control systems in Discussion to add a discussion of how the waypoint sequence approach is more appropriate in our particular case as “Additionally, the closed-loop control employs a waypoint sequence strategy, which is particularly suited to our microrobot’s discrete, stepwise locomotion and periodic feedback constraints, as it allows precise navigation with minimal angular adjustments compared to path-following methods that require dense reference points. This approach leverages the microrobot’s limited angular resolution and intermittent localization to optimize travel efficiency and robustness, as detailed in our analysis of locomotion characteristics, feedback timing, and control precision. Future improvements in real-time localization and actuation smoothness could enable more advanced path-following control.⁵⁷ Future integration of real-time localization and continuous control algorithms could enhance trajectory tracking and responsiveness.”

**Response to reviewer comments**

**Reviewer: 3**

The Authors have answer all my remaining questions. The article is ready for
publication.

**Response:** We sincerely thank the reviewer for the time and effort devoted to
evaluating our manuscript. We are grateful for the positive assessment and are pleased
that the revisions have addressed all remaining concerns.